# Continuous Parallel Relaxation for Finding Diverse Solutions in Combinatorial Optimization Problems

**Yuma Ichikawa**  *ichikawa.yuma@fujitsu.com*
*Fujitsu Limited, Kanagawa, Japan*

**Hiroaki Iwashita**  *iwashita.hiroak@fujitsu.com*
*Fujitsu Limited, Kanagawa, Japan*

**Reviewed on OpenReview:** *https://openreview.net/forum?id=ix33zd5zCw*

## Abstract

Finding the optimal solution is often the primary goal in combinatorial optimization (CO). However, real-world applications frequently require diverse solutions rather than a single optimum, particularly in two key scenarios. The first scenario occurs in real-world applications where strictly enforcing every constraint is neither necessary nor desirable. Allowing minor constraint violations can often lead to more cost-effective solutions. This is typically achieved by incorporating the constraints as penalty terms in the objective function, which requires careful tuning of penalty parameters. The second scenario involves cases where CO formulations tend to oversimplify complex real-world factors, such as domain knowledge, implicit trade-offs, or ethical considerations. To address these challenges, generating (i) *penalty-diversified* solutions by varying penalty intensities and (ii) *variation-diversified solutions* with distinct structural characteristics provides valuable insights, enabling practitioners to post-select the most suitable solution for their specific needs. However, efficiently discovering these diverse solutions is more challenging than finding a single optimal one. This study introduces **C**ontinual **P**arallel **R**elaxation **A**nnealing[1] (**CPRA**), a computationally efficient framework for unsupervised-learning (UL)-based CO solvers that generates diverse solutions within a single training run. CPRA leverages representation learning and parallelization to automatically discover shared representations, substantially accelerating the search for these diverse solutions. Numerical experiments demonstrate that CPRA outperforms existing UL-based solvers in generating these diverse solutions while significantly reducing computational costs.

## 1 Introduction

Constrained combinatorial optimization (CO) problems aim to find an optimal solution within a feasible space, a fundamental problem in various scientific and engineering applications (Papadimitriou & Steiglitz, 1998; Korte et al., 2011). However, real-world applications often require diverse solutions rather than a single optimal solution, particularly in two key scenarios.

The first scenario arises in real-world applications where strictly enforcing all constraints may neither be necessary nor desirable. In such cases, solutions that slightly violate certain constraints may be preferred if they result in significantly better cost performance. For example, soft deadlines in scheduling or minor rule relaxations in logistics may lead to more cost-effective and practically feasible solutions. A common approach to handling such flexibility is incorporating the constraints into the cost function as penalty terms, thereby transforming the original constrained optimization problem into an unconstrained one. While this formulation broadens the feasible solution space, it introduces the additional challenge of tuning penalty strengths to balance constraint satisfaction with cost minimization. To address this, a promising strategy

---

[1]The code is available at https://github.com/Yuma-Ichikawa/CPRA4CO.

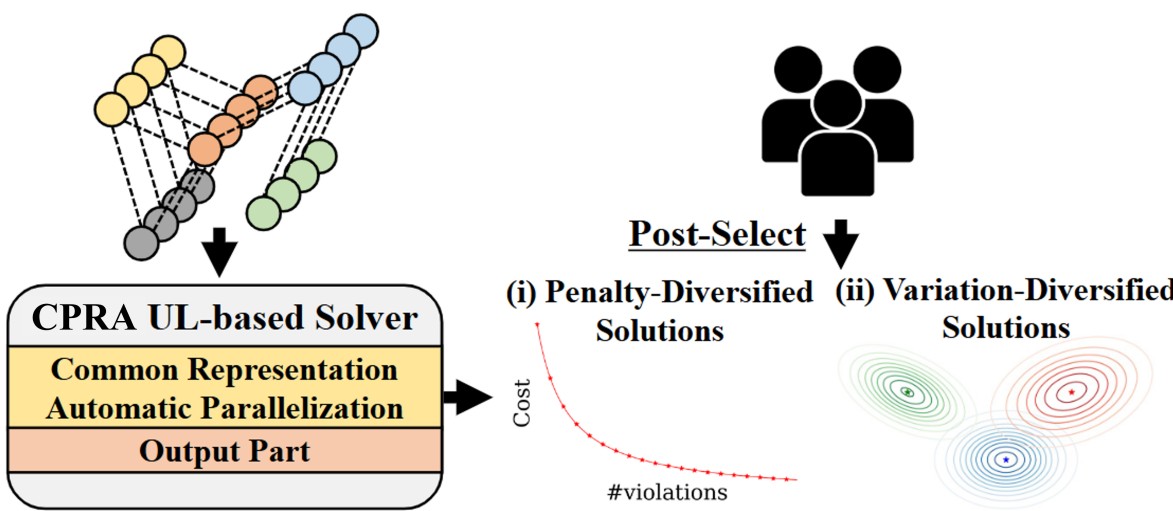

Figure 1: Overview of CPRA UL-based solver and post-processing for diverse solutions.

is to explore a set of solutions generated under varying penalty configurations, referred to as (i) *penalty-diversified solutions*. Such a solution set enables downstream decision-makers to select options based on contextual priorities, thereby enhancing both the robustness and practical utility of the solver's output, as illustrated in Fig. 1.

A second common scenario occurs in real-world settings where the mathematical formulation of a CO problem provides only a coarse approximation of the true decision-making environment. In practice, critical factors such as domain expertise, implicit trade-offs, and soft ethical or operational constraints are often too complex, dynamic, or subjective to be explicitly modeled. As a result, the optimal solution to a simplified formulation may not be practical or even desirable in real-world deployment. To bridge this gap, generating a diverse set of high-quality solutions that differ in structure or semantics is valuable, referred to as (ii) *variation-diversified solutions*. These solutions allow users to post-select the most suitable option, tailored to real-world complexities and adaptable to qualitative, contextual, or evolving criteria not captured by the original CO formulation, as illustrated in Fig. 1. Moreover, efficiently generating such diverse candidates is particularly important in high-stakes or large-scale domains, such as healthcare logistics, intelligent tutoring systems, and automated content generation, e.g., game level design (Zhang et al., 2020), where distinct feasible solutions may reflect differing operational priorities or ethical considerations. However, discovering such diverse solutions efficiently is considerably more challenging than identifying a single optimal solution.

To address these challenges, we propose the **C**ontinuous **P**arallel **R**elaxation **A**nnealing (CPRA) method, designed for unsupervised learning (UL)-based CO solvers (Schuetz et al., 2022a; Karalias & Loukas, 2020). CPRA enables the efficient discovery of both (i) penalty-diversified and (ii) variation-diversified solutions in a single training run, requiring only a minimal architectural change: extending the output layer from generating a single solution to producing multiple diverse solutions, while keeping the intermediate network structure unchanged. This modification offers substantial practical benefits by leveraging shared representations across problem instances. The resulting bottleneck structure promotes generalization and scalability, eliminating the need to train $S$ separate models iteratively. Furthermore, CPRA incorporates a controllable diversity mechanism for (ii) generating variation-diversified solutions, formulated as a continuous relaxation of the max-sum Hamming distance. Unlike the max-sum Hamming distance defined in binary space, which incurs a quadratic computational cost concerning the number of solutions, our approach reduces this to linear complexity, significantly improving the efficiency of both forward and backward passes.

Numerical experiments demonstrate the effectiveness of our approach on several benchmark CO problems. CPRA enables the generation of diverse solutions while maintaining comparable model size and runtime to UL-based solvers that produce only a single solution. Additionally, CPRA improves search performance by

leveraging shared representations across problem instances, resulting in higher-quality solutions than those produced by existing UL-based solvers and greedy algorithms.

## 2  Related work

Although diverse solutions can be generated through various paradigms beyond UL-based CO solvers, existing methods often differ significantly in their learning frameworks, algorithmic foundations, and architectural complexity (Grinsztajn et al., 2023; Choo et al., 2022; Li et al., 2023; Kwon et al., 2020; Kim et al., 2021; Xin et al., 2021; Bunel et al., 2018; Hottung et al., 2021; Chalumeau et al., 2023; Hottung et al., 2024). In many of these approaches, diversity is introduced as an auxiliary mechanism to support the discovery of a single optimal solution, rather than being pursued as a goal in its own right. Furthermore, these methods typically do not aim to generate (i) *penalty-diversified solutions*. In contrast, our work explicitly targets the efficient discovery of both (i) *penalty-diversified* and (ii) *variation-diversified* solutions, achieved through minimal modifications to existing UL-based combinatorial optimization solvers such as PI-GNN (Schuetz et al., 2022a) and CRA-PI-GNN (Ichikawa, 2024). Our primary goal is to demonstrate how a simple architectural extension can unlock broader solution diversity while maintaining practical runtime efficiency. Therefore, an exhaustive empirical comparison with all alternative paradigms falls outside the scope of this work. Penalty-diversified solutions can be understood as the result of treating cost and constraint violation as two distinct objectives and combining them through a weighted sum. This perspective naturally connects our framework with Pareto front learning methods in multi-objective optimization, notably Pareto hyper networks (Navon et al., 2021), preference-conditioned Pareto set learners (Lin et al., 2022), and the survey by Chen et al. (2025). Although these studies provide techniques to approximate the entire set of non-dominated trade-offs, a comprehensive empirical comparison with our approach remains an important avenue for future work.

**Diverse Solution Acquisition Without Neural CO Solvers.**  For penalty-diversified solutions, a straightforward baseline is to solve the same problem multiple times with different penalty coefficients, either in parallel, given sufficient CPU cores, or sequentially. However, both approaches are computationally intensive and scale poorly to large instance batches or real-time deployment scenarios. Our method overcomes this limitation by leveraging GPU parallelism to simultaneously generate solutions across multiple penalty regimes, achieving diversity with a runtime comparable to that of solving a single instance. Obtaining variation-diversified solutions is even more challenging. A common approach is to post-select solutions that maximize some diversity metric, such as the Hamming distance (Fernau et al., 2019). This line of research spans graph algorithms (Baste et al., 2019; 2022; Hanaka et al., 2021), constraint programming (Hebrard et al., 2005; Petit & Trapp, 2015), and mathematical programming (Danna et al., 2007; Danna & Woodruff, 2009; Petit & Trapp, 2019), and is often categorized into two types: (1) offline approaches that generate and filter a large pool of solutions, and (2) online approaches that incrementally construct diverse solutions. However, both methods face significant drawbacks. Offline approaches often produce redundant solutions and incur high memory and runtime costs. Online approaches, while more targeted, are inherently sequential, difficult to parallelize, and susceptible to local optima. Neither approach naturally scales to GPU-based parallel environments. In contrast, our method introduces a GPU-accelerated strategy for producing diverse high-quality solutions in a single forward pass. This capability is especially valuable in real-world, large-scale, or high-stakes settings such as scheduling, planning, or recommendation, where presenting multiple viable options is more useful than committing to a single solution and where computational efficiency is critical.

## 3  Notation.

We use the shorthand expression $[N] = \{1, 2, \ldots, N\}$, $N \in \mathbb{N}$. $I_N \in \mathbb{R}^{N \times N}$ represents an identity matrix of size $N \times N$. Here, $\mathbf{1}_N$ and $\mathbf{0}_N$ represent the all-ones vector and all-zeros vector in $\mathbb{R}^N$, respectively. $G(V, E)$ represents an undirected graph, where $V$ is the set of nodes and $E \subseteq V \times V$ is the set of edges. For a graph $G(V, E)$, $A$ denote the adjacency matrix with $A_{ij} = 0$ if an edge $(i, j)$ does not exist and $A_{ij} > 0$ if an edge connects $i$ and $j$. For a sequence $\{a_k \mid a_k \in \mathbb{R}\}_{k=1}^{K}$, the empirical variance is de-

fined as $\mathbb{VAR}[\{a_k\}_{k=1}^K] = \sum_{k=1}^K (a_k - \sum_{k'=1}^K a_{k'}/K)^2/K$, and the empirical standard deviation is given by $\mathbb{STD}[\{a_k\}_{k=1}^K] = (\mathbb{VAR}[\{a_k\}_{k=1}^K])^{1/2}$. For binary vectors $\boldsymbol{a}, \boldsymbol{b} \in \{0,1\}^N$, we define the Hamming distance as $d_H(\boldsymbol{a}, \boldsymbol{b}) = \sum_{i=1}^N \mathbf{1}[a_i \neq b_i]$ where $\mathbf{1}[\cdot]$ denotes the indicator function.

## 4 Background

### 4.1 Combinatorial Optimization (CO)

Constrained CO problems are defined as follows:

$$\min_{\boldsymbol{x} \in \mathcal{X}(C)} f(\boldsymbol{x}; C), \quad \mathcal{X}(C) = \left\{ \boldsymbol{x} \in \{0,1\}^N \ \middle| \ \begin{array}{ll} g_i(\boldsymbol{x}; C) \leq 0, & \forall i \in [I], \\ h_j(\boldsymbol{x}; C) = 0 & \forall j \in [J] \end{array} \right\}, \quad I, J \in \mathbb{N},$$

where $C \in \mathcal{C}$ denotes instance parameters, such as a graph $G = (V, E)$, where $\mathcal{C}$ denotes the set of all possible instances. The binary vector $\boldsymbol{x} = (x_i)_{1 \leq i \leq N} \in \{0,1\}^N$ is the decision variable to be optimized, and $\mathcal{X}(C)$ denotes the feasible solution space. $f : \mathcal{X} \times \mathcal{C} \to \mathbb{R}$ denotes the cost function and, for all $i \in [I]$ and $j \in [J]$, $g_i : \mathcal{X} \times \mathcal{C} \to \mathbb{R}$ and $h_j : \mathcal{X} \times \mathcal{C} \to \mathbb{R}$ denote constraints. In practical scenarios, constrained CO problems are often converted into unconstrained CO problems using the penalty method:

$$\min_{\boldsymbol{x} \in \{0,1\}^N} l(\boldsymbol{x}; C, \boldsymbol{\lambda}), \quad l(\boldsymbol{x}; C, \boldsymbol{\lambda}) \triangleq f(\boldsymbol{x}; C) + \sum_{i=1}^{I+J} \lambda_i v_i(\boldsymbol{x}; C).$$

where, for all $i \in [I + J]$, $v_i : \{0,1\}^N \times \mathcal{C} \to \mathbb{R}$ is the penalty term that increases when constraints are violated. For example, the penalty term is defined as follows:

$$\forall i \in [I], \ v_i(\boldsymbol{x}; C) = \max(0, g_i(\boldsymbol{x}; C)), \quad \forall j \in [J], \ v_j(\boldsymbol{x}; C) = (h_j(\boldsymbol{x}; C))^2$$

and $\boldsymbol{\lambda} = (\lambda_i)_{1 \leq i \leq I+J} \in \mathbb{R}_+^{I+J}$ represents the penalty parameters that balance satisfying the constraints and optimizing the cost function. Tuning these penalty parameters $\boldsymbol{\lambda}$ to obtain the desired solutions is a challenging and time-consuming task. This process often requires solving the problem multiple times while iteratively adjusting the penalty parameters $\boldsymbol{\lambda}$ until an acceptable solution is obtained.

### 4.2 Continuous Relaxation and UL-based Solvers

The continuous relaxation strategy reformulate a CO problem by converting discrete variables into continuous ones as follows:

$$\min_{\boldsymbol{p} \in [0,1]^N} \hat{l}(\boldsymbol{p}; C, \boldsymbol{\lambda}), \quad \hat{l}(\boldsymbol{p}; C, \boldsymbol{\lambda}) \triangleq \hat{f}(\boldsymbol{p}; C) + \sum_{i=1}^{I+J} \lambda_i \hat{v}_i(\boldsymbol{p}; C),$$

where $\boldsymbol{p} = (p_i)_{1 \leq i \leq N} \in [0,1]^N$ denotes relaxed continuous variables, i.e., each binary variable $x_i \in \{0,1\}$ is relaxed to a continuous one $p_i \in [0,1]$, and $\hat{f} : [0,1]^N \times \mathcal{C} \to \mathbb{R}$ is the relaxation of $f$, satisfying $\hat{f}(\boldsymbol{x}; C) = f(\boldsymbol{x}; C)$ for any $\boldsymbol{x} \in \{0,1\}^N$. The relation between the constraint $v_i$ and its relaxation $\hat{v}_i$ is similar for $i \in [I + J]$, i.e., $\forall i \in [I + J], \ \hat{v}_i(\boldsymbol{x}; C) = v_i(\boldsymbol{x}; C)$ for any $\boldsymbol{x} \in \{0,1\}^N$.

UL-based solvers employ this continuous relaxation strategy for training neural networks (NNs) (Wang et al., 2022; Schuetz et al., 2022a; Karalias & Loukas, 2020; Ichikawa, 2024). The relaxed continuous variables are parameterized by $\boldsymbol{\theta}$ as $\boldsymbol{p_\theta} \in [0,1]^N$ and optimized by directly minimizing the following loss function:

$$\hat{l}(\boldsymbol{\theta}; C, \boldsymbol{\lambda}) \triangleq \hat{f}(\boldsymbol{p_\theta}(C); C) + \sum_{i=1}^{I+J} \lambda_i \hat{v}_i(\boldsymbol{p_\theta}(C); C). \tag{1}$$

After training, the relaxed solution $\boldsymbol{p_\theta}$ is converted into discrete variables by rounding $\boldsymbol{p_\theta}$ using a threshold (Schuetz et al., 2022a) or by applying a greedy method (Wang et al., 2022). Two types of schemes have been developed based on this framework.

**(I) Learning Generalized Heuristics from History/Data.** One approach, proposed by Karalias & Loukas (2020), seeks to automatically learn commonly effective heuristics from historical dataset instances $\mathcal{D} = \{C_\mu\}_{\mu=1}^P$ and then apply these learned heuristics to a new instance $C^*$ via inference. Specifically, given a set of training instances, independently and identically distributed from a distribution $P(C)$, the objective is to minimize the average loss function $\min_\theta \sum_{\mu=1}^P l(\theta; C_\mu, \lambda)$. However, this method does not guarantee high-quality performance for a test instance $C^*$. Even if the training instances $\mathcal{D}$ are abundant and the test instance $C$ is drawn from the same distribution $P(C)$, achieving a low average performance $\mathbb{E}_{C \sim P(C)}[\hat{l}(\theta; C)]$ does not necessarily guarantee a low $l(\theta; C)$ for a specific $C$. To address this issue, Wang & Li (2023) introduced a meta-learning approach where NNs aim to provide good initialization for new instances.

**(II) Learning Effective Heuristics on Specific Single Instance.** Another approach, referred to as the physics-inspired graph neural networks (PI-GNN) solver (Schuetz et al., 2022a;b), automatically learns instance-specific heuristics for a given single instance using the instance parameter $C$ by directly employing Eq. (1). This approach has been applied to CO problems on graphs, i.e., $C = G(V, E)$, using graph neural networks (GNN) to model the relaxed variables $p_\theta(G)$. An $L$-layered GNN is trained to directly minimize $\hat{l}(\theta; C, \lambda)$ in Eq. (1), taking as input a graph $G$ along with node embedding vectors and producing the relaxed solution $p_\theta(G) \in [0, 1]^N$. A detailed description of GNNs can be found in Appendix B.1. Note that this setting is applicable even when the training dataset $\mathcal{D}$ is difficult to obtain. However, learning to minimize Eq. (1) for a single instance can be time-consuming than the inference process. Nonetheless, for large-scale problems, it has demonstrated superiority over other solvers in terms of both time and solution performance (Schuetz et al., 2022a;b; Ichikawa, 2024).

UL-based solvers face two practical issues: (I) "optimization issues", where they tend to get stuck in local optima, and (II) "rounding issues", which arise when an artificial post-learning rounding process is needed to map solutions from the continuous space back to the original discrete space, undermining the robustness of the results. To address the first issue, Ichikawa (2024); Ichikawa & Arai (2025) proposed annealing schemes to escape local optima by introducing the following entropy term $s(\theta; C)$:

$$\hat{r}(\theta; C, \lambda, \gamma) = \hat{l}(p_\theta(C); C, \lambda) + \gamma s(p_\theta(C)), \quad s(p_\theta(C)) = \sum_{i=1}^N \left\{ (2p_{\theta,i}(C) - 1)^\alpha - 1 \right\}, \quad \alpha \in \{2n \mid n \in \mathbb{N}\}, \quad (2)$$

where $\gamma \in \mathbb{R}$ denotes a penalty parameter. They anneal the penalty parameter from positive $\gamma > 0$ to $\gamma \approx 0$ to smooth the non-convexity of the objective function $\hat{l}(\theta; C, \lambda)$ similar to simulated annealing (Kirkpatrick et al., 1983). To address the second issue, Ichikawa (2024) further annealed the entropy term to $\gamma \leq 0$ until the entropy term approaches zero, i.e., $s(\theta, C) \approx 0$, enforcing the relaxed variable to take on discrete values and further smoothing the continuous loss landscape for original discrete solutions. This method is referred to as **C**ontinuous **R**elaxation **A**nnealing (**CRA**), and the solver that applies the CRA to the PI-GNN solver is referred to as CRA-PI-GNN solver.

## 5 Continuous Parallel Relaxation Annealing for Diverse Solutions

We propose an extension of CRA, termed **C**ontinuous **T**ensor **R**elaxation (**CPRA**), which enables UL-based solvers to efficiently handle multiple problem instances within a single training run. Beyond this core advancement, we demonstrate how CPRA can be effectively tailored to discover both *penalty-diversified solutions* and *variation-diversified solutions*.

### 5.1 Continuous Parallel Relaxation (CPRA)

Let us consider solving multiple instances $\mathcal{C}_S = \{C_s \mid C_s \in \mathcal{C}\}_{1 \leq s \leq S}$ with different penalty parameters $\Lambda_S = \{\lambda_s\}_{1 \leq s \leq S}$ simultaneously. To handle these instances, we relax a binary vector $x \in \{0, 1\}^N$ into an augmented continual matrix $P \in [0, 1]^{N \times S}$ that is trained via minimizing the following loss function:

$$\hat{R}(P; \mathcal{C}_S, \Lambda_S, \gamma) = \sum_{s=1}^S \hat{l}(P_{:s}; C_s, \lambda_s) + \gamma S(P), \quad S(P) \triangleq \sum_{i=1}^N \sum_{s=1}^S (1 - (2P_{is} - 1)^\alpha),$$

where $P_{:s} \in [0,1]^N$ denotes $s$-the column in $P$, i.e. $P = (P_{:s})_{1 \le s \le S} \in [0,1]^{N \times S}$. Optimizing $\hat{R}$ drives each column $P_{:s}$ to minimize its respective objective function $\hat{l}(P_{:s}; C_s, \boldsymbol{\lambda}_s)$. Additionally, we also generalize the entropy term $s(\boldsymbol{p})$ in Eq. (2) into $S(P)$ for this augmented higher-order array. Specifically, the following theorem holds.

**Theorem 5.1.** *Under the assumption that for all $s \in [S]$, each objective function $\hat{l}(P_{:s}; C_s, \boldsymbol{\lambda}_s)$ remains bounded on $[0,1]^N$, each column solutions $P_{:s}^*$ such that $P^* \in \arg\min_P \hat{R}(P; \mathcal{C}_S, \Lambda_S, \gamma)$ converges to the corresponding discrete optimal $\boldsymbol{x}^* \in \arg\min_{\boldsymbol{x}} l(\boldsymbol{x}; C_s, \boldsymbol{\lambda}_s)$ as $\gamma \to +\infty$. Furthermore, as $\gamma \to -\infty$, the loss function $\hat{R}(P; \mathcal{C}_S, \Lambda_S)$ becomes convex and admits a unique half-integral solution $\mathbf{1}_N \mathbf{1}_N^\top / 2 = \arg\min_P \hat{R}(P; \mathcal{C}_S, \Lambda_S, \gamma)$.*

A detailed proof of Theorem 5.1 is available in Appendix A.1. The relaxation approach naturally extends to higher-order arrays, $P \in [0,1]^{N \times S_1 \times \cdots}$, potentially enabling more powerful GPU-based parallelization. A comprehensive exploration of such higher-dimensional implementations remains an exciting avenue for future research. For UL-based solvers, we parameterize the soft higher-order array $P$ as $P_{\boldsymbol{\theta}}$, leading to

$$\hat{R}(\boldsymbol{\theta}; \mathcal{C}_S, \Lambda_S, \gamma) = \sum_{s=1}^{S} \hat{l}(P_{\boldsymbol{\theta},:s}(\mathcal{C}_S); C_s, \boldsymbol{\lambda}_s) + \gamma S(P_{\boldsymbol{\theta}}(\mathcal{C}_S)), \;\; S(P_{\boldsymbol{\theta}}(\mathcal{C}_S)) \triangleq \sum_{i=1}^{N} \sum_{s=1}^{S} (1 - (2P_{\boldsymbol{\theta},is}(\mathcal{C}_S) - 1)^\alpha), \quad (3)$$

where $\gamma$ is also annealed from a positive to a negative value as in CRA-PI-GNN solver in Section 4. Following the UL-based solvers (Karalias & Loukas, 2020; Schuetz et al., 2022b; Ichikawa, 2024), we encode $P_\theta$ via a GNN-based architecture. This study refer to the solver that applies CPRA to PI-GNN solver as CPRA-PI-GNN solver.

**Simple Architectural Modifications for Efficient Learning and Shared Representation Learning.** In this study, we employ a specialized GNN-based architecture that builds upon the core designs of PI-GNN (Schuetz et al., 2022a) and CRA-PI-GNN (Ichikawa, 2024) to simultaneously address multiple problem instances, as defined in Eq. (3). Unlike these solvers that generate a single solution per instance, CPRA-PI-GNN handles $S$ instances in parallel by expanding the final-layer node embedding dimension from 1 to $S$, as illustrated in Fig. 1. As a result, the number of parameters increases linearly only in the output layer, while the rest of the architecture remains unchanged. This design is both memory- and computation-efficient, as the overall network size and training time remain comparable to solving a single instance.

Furthermore, by keeping the network unchanged apart from the output layer, CPRA-PI-GNN solver naturally encourages the model to learn compact, shared representations across multiple instances, functioning similarly to a bottleneck in an autoencoder, as illustrated in Fig. 1. This simple architectural choice improves solution quality by leveraging the learned representations. Indeed, numerical experiments demonstrate that this shared representation learning strategy yields better results than single-instance solvers such as PI-GNN and CRA-PI-GNN. Additional results in Appendix D.5 further show that CPRA-PI-GNN solves multiple similar problems more efficiently and effectively than CRA-PI-GNN. To further reduce computational cost, we adopt a two-stage learning process that leverages shared representation learning: first training on a smaller, representative subset $S' \subset S$, then fine-tuning only the final-layer embeddings when scaling to the full set $S$. This approach offers an efficient way to extend the model to larger problem sets.

## 5.2 CPRA for Finding Penalty-Diversified Solutions

To find *penalty-diversified solutions*, we aim to minimize the following loss function for a problem instance $C$, which is a special case of Eq. (3):

$$\hat{R}(\boldsymbol{\theta}; C, \Lambda_S, \gamma) = \sum_{s=1}^{S} \hat{l}(P_{\boldsymbol{\theta},:s}(C); C, \boldsymbol{\lambda}_s) + \gamma S(P_{\boldsymbol{\theta}}(C)). \quad (4)$$

By solving this optimization problem, each column $P_{\theta,:s}(C_s)$, for all $s \in [S]$, corresponds to the optimal solution for the penalty parameter $\boldsymbol{\lambda}_s$. For penalty-diversified solutions, the variation in each $s$ is primarily restricted to the penalty coefficient, leading to a strong correlation among instances.

### 5.3 CPRA for Finding Variation-Diversified Solutions

To explore *variation-diversified solutions* for a single instance $C$ with a penalty parameter $\boldsymbol{\lambda}$, we introduce a diversity penalty into Eq. (3) as follows:

$$\hat{R}(\boldsymbol{\theta}; C, \boldsymbol{\lambda}, \gamma, \nu) = \sum_{s=1}^{S} \hat{l}(P_{\boldsymbol{\theta},:s}(C); C, \boldsymbol{\lambda}) + \gamma S(P_{\boldsymbol{\theta}}(C)) + \nu \Psi(P_{\boldsymbol{\theta}}(C)),$$

$$\Psi(P_{\boldsymbol{\theta}}(C)) = -S \sum_{i=1}^{N} \mathbb{STD}\left[\{P_{\boldsymbol{\theta},is}(C)\}_{1 \le s \le S}\right], \quad (5)$$

where $\Psi(P_{\boldsymbol{\theta}}(C))$ serves as a constraint term that promotes diversity in each column $P_{\boldsymbol{\theta},:s}(C)$, and $\nu$ is the parameter controlling the strength of this constraints. Setting $\nu = 0$ in Eq. (5) is nearly equivalent to solving the same CO problem with different initial conditions.

As shown in Proposition 5.2, the diversity term $\Psi(P_{\boldsymbol{\theta}}(C))$ can be interpreted as a natural relaxation of the max-sum Hamming distance, a widely used diversity metric in combinatorial optimization (Fomin et al., 2020; 2023; Baste et al., 2022; 2019). Specifically, for a set of binary sequences $\{\boldsymbol{x}^{(s)}\}_{s=1}^{S}$ with $\boldsymbol{x}^{(s)} \in \{0,1\}^N$, the following holds:

**Proposition 5.2.** *For a set of binary sequences $\{\boldsymbol{x}^{(s)}\}_{s=1}^{S}$, $\forall s$, $\boldsymbol{x}^{(s)} \in \{0,1\}^N$, the following equality holds:*

$$S^2 \sum_{i=1}^{N} \mathbb{VAR}\left[\left\{x_i^{(s)}\right\}_{1 \le s \le S}\right] = \sum_{s<l} d_H(\boldsymbol{x}^{(s)}, \boldsymbol{x}^{(l)}), \quad (6)$$

*where the right-hand side of Eq.* (6) *represents the max-sum Hamming distance.*

The detailed proof can be found in Appendix A.2. This proposition shows that the proposed diversity penalty corresponds exactly to the max-sum Hamming distance when evaluated at the binary vertices of the hypercube $\{0,1\}^N$, which means the left hand side of Eq. 6 is a natural continuous relaxation of the discrete diversity measure onto the continuous domain $[0,1]^N$.

This relaxation yields a significant computational advantage. Directly computing the max-sum Hamming distance involves evaluating all $\binom{S}{2} = \mathcal{O}(S^2)$ pairwise distances between solutions, which becomes computationally expensive as $S$ increases. In contrast, the relaxed diversity term $\Psi(\cdot)$ computes per-coordinate variance using simple summary statistics, which can be evaluated in $\mathcal{O}(S)$ time. This efficiency holds for both forward and backward passes, approaching well-suited to gradient-based optimization. To ensure consistent scale and sensitivity with the other loss components in Eq. (5), we normalize the diversity term using the sample standard deviation.

## 6 Experiments

This section evaluate the effectiveness of CPRA-PI-GNN solver in discovering penalty-diversified and variation-diversified solutions across three CO problems: the maximum independent set (MIS), maximum cut (MaxCut), diverse bipartite matching (DBM) problems. Their objective functions are summarized in Table 1 in Appendix C.3; For a detailed explanation, refer to Appendix C.3.

### 6.1 Settings

**Baseline.** Our baseline include results from executing a greedy algorithms, PI-GNN solver (Schuetz et al., 2022a) and CRA-PI-GNN solver (Ichikawa, 2024) multiple times. These solvers are executed multiple times using different penalty parameters for penalty-diversified solutions and different random seeds for variation-diversified solutions, allowing us to assess the search efficiency for both types of diversified solutions. For the MIS problem, we employ a random greedy search implemented by `NetworkX`, and for the MaxCut problem, we use a random greedy search implemented by Mehta (2019). Although some online heuristics exist for

|  | MIS ($d = 5$) | | |
|---|---|---|---|
| Method {#Runs} | #Params | Time (s) | ApR* |
| PI-GNN {20} | 5,022,865×20 | 13,189±60 | 0.883±0.002 |
| CRA {20} | 5,022,865×20 | 14,400±42 | **0.961±0.002** |
| CPRA {1} | **5,083,076** | **1,194±8** | 0.934±0.002 |

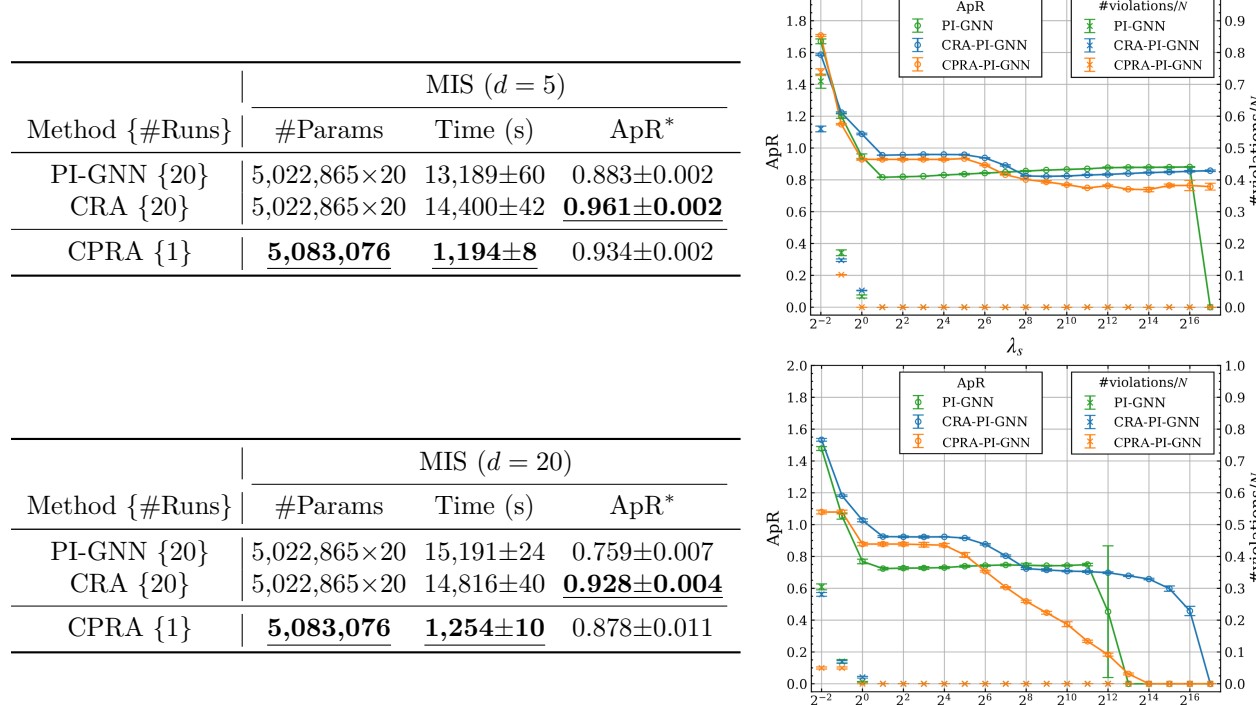

|  | MIS ($d = 20$) | | |
|---|---|---|---|
| Method {#Runs} | #Params | Time (s) | ApR* |
| PI-GNN {20} | 5,022,865×20 | 15,191±24 | 0.759±0.007 |
| CRA {20} | 5,022,865×20 | 14,816±40 | **0.928±0.004** |
| CPRA {1} | **5,083,076** | **1,254±10** | 0.878±0.011 |

Figure 2: (Left Table) shows runtime (Time), number of parameters (#Params), and maximum ApR (ApR*) for each method. (Right Figure) shows ApRs across different penalty parameters $\Lambda_s$. Error represent the standard deviations of 5 random seeds. CPRA-PI-GNN solver can find penalty-diversified solutions in a single run with a comparable #Params and runtime to UL-based solvers that output a single solution.

exploring variation-diversified solutions by generating solutions that are distant from those already obtained, we do not include these methods as benchmarks due to their inefficient GPU utilization and poor scalability to large problems. We measure the runtime $t$ of each execution, from model training to the final output.

**Implementation.** This numerical experiment aims to validate that CPRA can generate penalty-diversified and variation-diversified solutions while maintaining a comparable number of parameters and runtime to UL-based solvers, which produce a single solution, as described in Section 5. Therefore, in our experiments, the CPRA-PI-GNN solver utilizes the same network architecture as the PI-GNN (Schuetz et al., 2022a) and CRA-PI-GNN (Ichikawa, 2024) solvers, except for the output size of the final layer as discussed in Section 5. We use `GraphSage`, implemented with the Deep Graph Library (Wang et al., 2019). The detailed architectures of these GNNs are provided in Appendix C.1. We employ the AdamW (Kingma & Ba, 2014) optimizer with a learning rate of $\eta = 10^{-4}$ and a weight decay of $10^{-2}$. The GNNs are trained for up to $5 \times 10^4$ epochs with early stopping, which monitors the summarized loss function $\sum_{s=1}^{S} \hat{l}(P_{:,s})$ and the entropy term $\Phi(P; \gamma, \alpha)$, using a tolerance of $10^{-5}$ and patience of $10^3$ epochs. Further details are provided in Appendix C.2. We set the initial scheduling value to $\gamma(0) = -20$ for the MIS and DBM problems and $\gamma(0) = -6$ for the MaxCut problems, using the same scheduling rate $\varepsilon = 10^{-3}$ and curvature rate $\alpha = 2$ in Eq. (3).

**Evaluation Metrics.** Following the metric of Wang & Li (2023), we use the approximation rate (ApR) for all experiments, defined as $\text{ApR} = f(\boldsymbol{x}; C)/f(\boldsymbol{x}^*; C)$, where $\boldsymbol{x}^*$ represents the optimal solutions. For MIS, these optimal solutions set to the theoretical results (Barbier et al., 2013), for DBM problems, they are identified using Gurobi 10.0.1 solver with default settings, and for MaxCut problems, they are the best-known solutions. To evaluate the quality of penalty-diversified solutions, we compute $\text{ApR}^* = \max_{s \in [S]}(\text{ApR}(\boldsymbol{x}_s))$ as a function of the parallel number $S$ in Eq. (3). To evaluate the quality of variation-diversified solutions, we compute the average ApR, defined as $\overline{\text{ApR}} = \sum_{s=1}^{S} \text{ApR}_s / S$, and introduce a diversity score (DScore) for

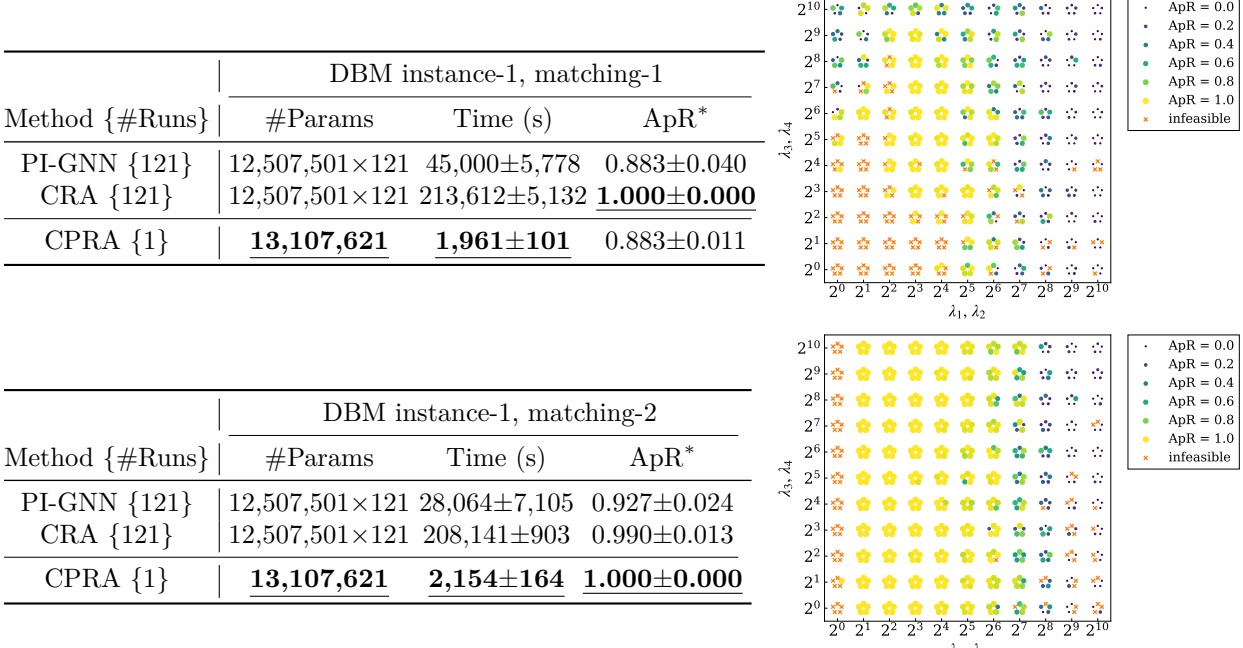

| Method {#Runs} | DBM instance-1, matching-1 | | |
|---|---|---|---|
| | #Params | Time (s) | ApR$^*$ |
| PI-GNN {121} | 12,507,501×121 | 45,000±5,778 | 0.883±0.040 |
| CRA {121} | 12,507,501×121 | 213,612±5,132 | **1.000±0.000** |
| CPRA {1} | **13,107,621** | **1,961±101** | 0.883±0.011 |

| Method {#Runs} | DBM instance-1, matching-2 | | |
|---|---|---|---|
| | #Params | Time (s) | ApR$^*$ |
| PI-GNN {121} | 12,507,501×121 | 28,064±7,105 | 0.927±0.024 |
| CRA {121} | 12,507,501×121 | 208,141±903 | 0.990±0.013 |
| CPRA {1} | **13,107,621** | **2,154±164** | **1.000±0.000** |

Figure 3: (Left Table) shows runtime (Time), number of parameters (#Params), and maximum ApR (ApR$^*$) for each method, with errors representing the standard deviations of 5 random seeds. (Right Figure) shows ApRs, where each point represents the results from 5 random seed across various penalty parameters $\Lambda_S = \{\boldsymbol{\lambda}_s = (\lambda_a, \lambda_a, \lambda_b, \lambda_b) \mid \lambda_a, \lambda_b \in \{2^s \mid s = 0, \ldots, 10\}\}$. CPRA-PI-GNN solver is capable of finding penalty-diversified solutions in a single run, with a comparable number of parameters and runtime to those of UL-based solvers.

the bit sequences $\{\boldsymbol{x}_s\}_{s=1}^S$:

$$\text{DScore}(\{\boldsymbol{x}_s\}_{s=1}^S) = \frac{2}{NS(S-1)} \sum_{s<l} d_H(\boldsymbol{x}_s, \boldsymbol{x}_l)$$

A higher DScore indicates greater variation among solutions. A desirable variation-diversified solution should exhibit both high-quality solutions and a diverse set of solutions with distinct characteristics. Thus, solutions with higher values of both average ApR and DScore are more desirable.

## 6.2 Finding Penalty-Diversified Solutions

**MIS Problems.** First, we compare the performacne of CPRA-PI-GNN on MIS problems in RRGs, $G(V, E)$, with $|V| = 10,000$ nodes and the node degree of 5 and 20. CPRA-PI-GNN solver run using Eq. (4), with a set of penalty parameters, $\Lambda_S = \{2^{s-3} \mid s = 1, \ldots, 20\}$. CRA-PI-GNN and PI-GNN solver run multiple times for each penalty parameter $\lambda_s \in \Lambda_S$. Fig. 2 (Right) shows the ApR as a function of penalty parameters $\lambda_s \in \Lambda_S$. Across all penalty parameters, from $2^{-2}$ to $2^{16}$, CPRA-PI-GNN solver performs on par with or slightly underperforms CRA-PI-GNN solver. Table in Fig. 2 shows the runtime and number of paramers (#Params) for CPRA-PI-GNN solver at $S = 20$, compared to the total runtime and #Params for $S$ runs of PI-GNN and CRA-PI-GNN solvers. These result indicate that CPRA-PI-GNN solver can find penalty-diversified solutions with a comparable number of parameters and runtime to UL-based solvers that output a single solution. For a more detailed discussion on the dependence of the runtime and the #params for number of shot $S$, refer to Appendix D.1.

**DBM Problems.** We next demonstrate the effectiveness of CPRA-PI-GNN solver for DBM problems, which serve as practical CO problems. We focus on the first of the 27 DBM instances; see Appendix

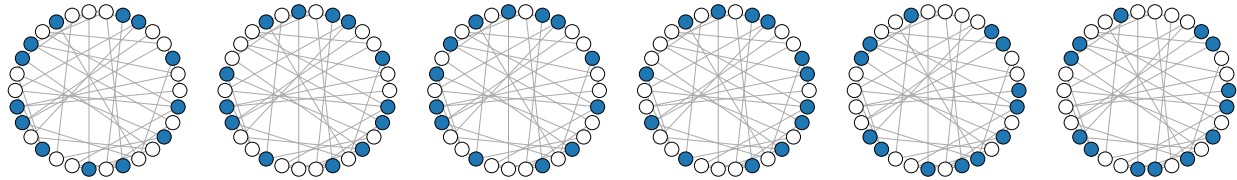

Figure 4: The obtained solutions by CPRA-PI-GNN solver for the MIS problem on a RRG with 30 nodes and the degree $d = 3$. Blue nodes represent the independent set.

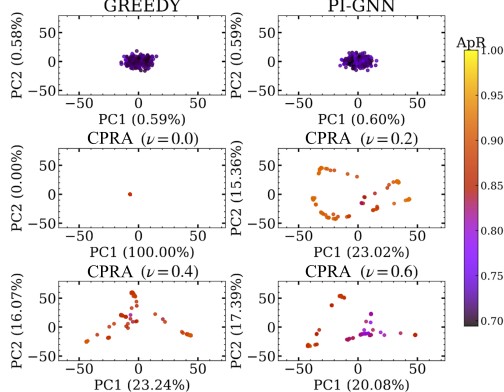

|  | MIS ($d = 20$) | | |
|---|---|---|---|
| Method {#Runs} | Time (s) | A$\bar{\text{p}}$R | DScore |
| Greedy {300} | **8** | 0.715 | 0.239 |
| PI-GNN {300} | 13,498 | 0.712 | 0.238 |
| CRA {300} | 15,136 | 0.923 | 0.248 |
| CPRA ($\nu = 0.0$) {1} | 95 | 0.873 | 0.019 |
| CPRA ($\nu = 0.2$) {1} | 154 | **0.936** | **0.260** |
| CPRA ($\nu = 0.4$) {1} | 154 | 0.900 | 0.251 |
| CPRA ($\nu = 0.6$) {1} | 149 | 0.852 | 0.257 |

Figure 5: (Left Table) shows runtime (Time), average ApR (A$\bar{\text{p}}$R), DScore for each method on MIS problems with a node degree $d = 20$. (Right Figure) shows the distribution of solutions in a 2-dimensional space using PCA with varying $\nu$.

D.6 for the results of the remaining instances. Given that $(\lambda_1, \lambda_2)$ and $(\lambda_3, \lambda_4)$ share similar properties, CPRA-PI-GNN run with a set of $S = 11 \times 11$ parameters on the a grid, $\Lambda_S = \{\boldsymbol{\lambda}_s = (\lambda_a, \lambda_a, \lambda_b, \lambda_b)\}$, where $\lambda_a, \lambda_b \in \{2^s \mid s = 0, \ldots, 10\}$. CRA-PI-GNN and PI-GNN solver run multiple times for each penalty parameter $\lambda_s \in \Lambda_S$ Fig. 3 (Right) shows that the ApR on the grid $\Lambda_S$ using the CPRA-PI-GNN solver identifies a desirable region where the ApR is nearly 1.0. Table in Fig. 3 demonstrates that CPRA-PI-GNN solver can find penalty-diversified solutions with a comparable number of parameters and runtime to UL-based solvers that output a single solution.

## 6.3 Finding Variation-Diversified Solutions

We next demonstrate that CPRA-PI-GNN solver can efficiently find variation-diversified solutions. Furthermore, we also show that the CPRA-PI-GNN solver enhances exploration capabilities and achieves higher-quality solutions.

**MIS Problems.** We first run CPRA-PI-GNN solver using Eq. (5) to find variation-diversified solutions for MIS problems on small-scaled RRGs with 30 nodes and the node degree set to 3. We set the parameter $\nu = 0.5$ and the number of shots tp $S = 100$ in Eq. (5). As shown in Fig. 4, CPRA-PI-GNN solver successfully obtain 6 solutions, each with 13 independent sets, which is the global optimum. We extend the investigation to large-scale RRG with 10,000 nodes and a node degree $d = 20$, which is known for its optimization challenges (Angelini & Ricci-Tersenghi, 2023). These experiments investigate how the quality of variation-diversified solutions depends on the parameter $\nu$, using a fixed number of shots $S = 300$. Fig. 5 (Right) shows a low dimensional visualization of the normalized solutions $\{P_{:s}\}_{s=1}^{300}$ using two-dimensional principal component analysis (PCA) mapping. The two principal components with the highest contribution rates are selected for different parameters $\nu = 0.0, 0.2, 0.4, 0.6$. These results indicate that increasing parameter $\nu$ leads to more diverse solutions, with the solution space becoming increasingly separated in the high-contribution region. Table in Fig. 5 measures the computation time, A$\bar{\text{p}}$R, and DScore when the parallel execution

|  | MaxCut G14 | | |
|---|---|---|---|
| Method {#Runs} | Time (s) | ApR̄ | DScore |
| Greedy {1,000} | **87** | 0.936 | 0.479 |
| PI-GNN {1,000} | 25,871 | 0.963 | 0.499 |
| CRA {1,000} | 48,639 | 0.988 | 0.499 |
| CPRA ($\nu = 0.0$) {1} | 144 | 0.977 | 0.497 |
| CPRA ($\nu = 0.4$) {1} | 138 | **0.991** | 0.501 |
| CPRA ($\nu = 0.8$) {1} | 141 | 0.989 | 0.501 |
| CPRA ($\nu = 1.2$) {1} | 147 | 0.985 | **0.502** |

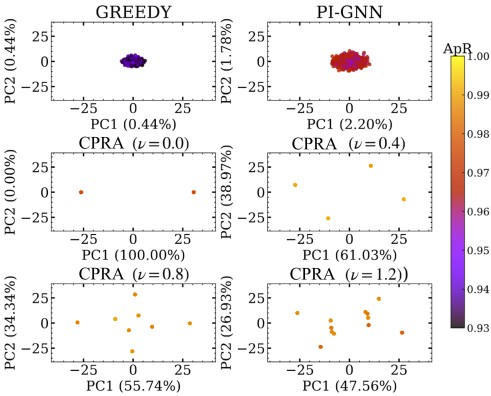

Figure 6: (Left Table) shows runtime (Time), average ApR (ApR̄), DScore for each method on MaxCut G14, with error representing the standard deviations of 5 random seeds. (Right Figure) shows the distribution of solutions in a 2-dimensional space using PCA with varying $\nu$.

number $S = 300$ using different random seeds. The results show that although the time of CPRA-PI-GNN solver takes longer than executing the greedy algorithm multiple times, both ApR and DScore reach their maximum at $\nu = 0.2$, yielding the highest quality variation-diversified solutions. Furthermore, increasing parameter $\nu$ enhances the exploration capability of GNN, leading to better solutions than those obtained by conventional PI-GNN and CRA-PI-GNN; see Appendix D.2.

**MaxCut Problems.** Next, we evaluate the ability to find variation-diversified solutions in the G14 instance of Gset, which primarily has four-clustered solution space. We set the number of shot $S = 1,000$ in Eq. (5). Fig. 6 (Right) demonstrates that CPRA-PI-GNN solver can capture four-clustered solutions beyond a certain value of $\nu$. Table in Fig. 6 measures the computation time, ApR̄, and DScore when the parallel execution number $S = 1,000$ is performed using different random seeds. The results show that although the runtime of CPRA-PI-GNN solver is slower compared to executing the greedy algorithm multiple times, ApR and DScore reach their maximum at $\nu = 0.4$ and $\nu = 1.2$, respectively. Additionally, similar to MIS problems, exploration enhancement is consistent across various instances of Gset. For further details; see Appendix D.3.

## 7 Conclusion

This study introduces the CPRA framework for UL-based solvers designed to efficiently find penalty-diversified and variation-diversified solutions within a single training process. Our numerical experiments demonstrate that CPRA can produce penalty-diversified and variation-diversified solutions while maintaining a comparable number of parameters and runtime to conventional UL-based solvers that generate only a single solution. This approach not only enhances the computational efficiency in finding these diversified solutions but also improves the search capabilities, leading to higher-quality solutions compared to existing UL-based solvers that find a single solution and greedy algorithms.

**Acknowledgments**

We would like to thank Hisanao Akima and Nobuo Namura. We also appreciate the constructive feedback provided by the anonymous reviewers and the action editor. This work was supported by Fujitsu Limited.

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

# A Derivations

## A.1 Proof of Theorem 3.1

Following the proof of Ichikawa (2024), we show Theorem 5.1 based on following three lemmas.

**Lemma A.1.** *For any even natural number $\alpha = 2, 4, \ldots$, the function $\phi(p) = 1 - (2p - 1)^\alpha$ defined on $[0, 1]$ achieves its maximum value of 1 when $p = 1/2$ and its minimum value of 0 when $p = 0$ or $p = 1$.*

*Proof.* The derivative of $\phi(p)$ relative to $p$ is $\phi'(p) = -2\alpha(2p - 1)$, which is zero when $p = 1/2$. This is a point where the function is maximized because the second derivative $\phi''(p) = -4\alpha \leq 0$. In addition, this function is concave and symmetric relative to $p = 1/2$ because $\alpha$ is an even natural number, i.e., $\phi(p) = \phi(1 - p)$, thereby achieving its minimum value of 0 when $p = 0$ or $p = 1$. $\square$

**Lemma A.2.** *For any even natural number $\alpha = 2, 4, \ldots$ and a matrix $P \in [0, 1]^{N \times S}$, if $\lambda \to +\infty$, minimizing the penalty term $\Phi(P; \gamma) = \gamma \sum_{s=1}^{S} \sum_{i=1}^{N} (1 - (2P_{is} - 1)^\alpha) = \gamma \sum_{s=1}^{S} \sum_{i=1}^{N} \phi(P_{is}; \alpha)$ enforces that the components of $P_{is}$ must be either 0 or 1 and, if $\gamma \to -\infty$, the penalty term enforces $P = \mathbf{1}_N \mathbf{1}_N^\top / 2$.*

*Proof.* From Lemma A.1, as $\gamma \to +\infty$, the case where $\phi(P_{is})$ becomes minimal occurs when, for each $i, s$, $p_{is} = 0$ or $p_i = 1$. In addition, as $\gamma \to -\infty$, the case where $\phi(p; \gamma)$ is minimized occurs when, for each $i$, $P_{is}$ reaches its maximum value with $P_{is} = 1/2$. $\square$

**Lemma A.3.** $\Phi(P; \gamma) = \gamma \sum_{s=1}^{S} \sum_{i=1}^{N} (1 - (2p_i - 1)^\alpha) = \gamma \sum_{s=1}^{S} \sum_{i=1}^{N} \phi(p_i; \alpha)$ *is concave when $\lambda$ is positive and is a convex function when $\lambda$ is negative.*

*Proof.* Note that $\Phi(P; \gamma) = \gamma \sum_{s=1}^{S} \sum_{i=1}^{N} \phi(P_{is}; \alpha) = \gamma \sum_{i=1}^{N} (1 - (2P_{is} - 1)^\alpha)$ is separable across its components $P_{is}$. Thus, it is sufficient to prove that each $\gamma \phi(P_{is}; \alpha)$ is concave or convex in $P_{is}$ because the sum of the concave or convex functions is also concave (and vice versa). Therefore, we consider the second derivative of $\gamma \phi_i(P_{is}; \alpha)$ with respect to $P_{is}$:

$$\gamma \frac{d^2 \phi_i(P_{is}; \alpha)}{dP_{is}^2} = -4\gamma\alpha$$

Here, if $\gamma > 0$, the second derivative is negative for all $p_i \in [0, 1]$, and this completes the proof that $\Phi(P; \gamma, \alpha)$ is a concave function when $\gamma$ is positive over the domain $\boldsymbol{p} \in [0, 1]^N$ $\square$

**Theorem A.4.** *Under the assumption that the objective function $\sum_s \hat{l}(P_{:s}; C_s, \boldsymbol{\lambda}_s)$ is bounded within the domain $[0, 1]^{N \times S}$, for any $S \in \mathbb{N}$, $C_s \in \mathcal{C}_S$ and $\boldsymbol{\lambda}_s \in \Lambda_S$, as $\gamma \to +\infty$, each column $P_{:s}^*$ of the soft solutions $P^* \in \arg\min_P \hat{R}(P; \mathcal{C}_S, \Lambda_S, \gamma)$ converges to the original solutions $\boldsymbol{x}^* \in \arg\min_{\boldsymbol{x}} l(\boldsymbol{x}; C_s, \boldsymbol{\lambda}_s)$. In addition, as $\gamma \to -\infty$, the loss function $\hat{R}(P; \mathcal{C}_S, \Lambda_S)$ becomes convex and the soft solution $\mathbf{1}_N \mathbf{1}_N^\top / 2 = \arg\min_P \hat{R}(P; \mathcal{C}_S, \Lambda_S, \gamma)$ is unique.*

*Proof.* As $\lambda \to +\infty$, the penalty term $\Phi(P; \boldsymbol{\lambda})$ dominates the loss function $\hat{R}(\boldsymbol{p}; C, \boldsymbol{\lambda}, \gamma)$. According to Lemma A.2, this penalty term forces the optimal solution $P^*$ to have components $p_{is}^*$ that are either 0 or 1 because any nonbinary value will result in an infinitely large penalty. This effectively restricts the feasible region to the vertices of the unit hypercube, which correspond to the binary vector in $\{0, 1\}^{NS}$. Thus, as $\lambda \to +\infty$, the solutions to the relaxed problem converge to $X = \arg\min_{X \in \{0,1\}^{N \times S}} R(X_{:s}; C_s, \boldsymbol{\lambda}_s)$. Futhermore, $\arg\min_{X \in \{0,1\}^{N \times S}} R(X_{:s}; C_s, \boldsymbol{\lambda}_s)$ is separable as $\sum_{s=1}^{S} \arg\min_{\boldsymbol{x} \in \{0,1\}^N} l(\boldsymbol{x}; C_s, \boldsymbol{\lambda}_s)$, which indicate that each columns $X_{:s}^* \in \arg\min_{\boldsymbol{x} \in \{0,1\}^N} l(\boldsymbol{x}; C_s, \boldsymbol{\lambda}_s)$. As $\lambda \to -\infty$, the penalty term $\Phi(\boldsymbol{p}; \alpha)$ also dominates the loss function $\hat{r}(\boldsymbol{p}; C, \boldsymbol{\lambda}, \gamma)$ and the $\hat{r}(\boldsymbol{p}; C, \boldsymbol{\lambda})$ convex function from Lemma A.3. According to Lemma A.2, this penalty term forces the optimal solution $P^* = \mathbf{1}_N \mathbf{1}_N^\top / 2$. $\square$

## A.2 Proof of Proposition 5.2

In this section, we derive the following Proposition.

**Proposition A.5.** *For binary sequences $\{\boldsymbol{x}_s\}_{s=1}^S$, $\forall s$, $\boldsymbol{x}_s \in \{0,1\}^N$, following equality holds*

$$S^2 \sum_{i=1}^{N} \mathbb{VAR}\left[\{\boldsymbol{x}_{s,i}\}_{1 \leq s \leq S}\right] = \sum_{s<l} d_H(\boldsymbol{x}_s, \boldsymbol{x}_l). \tag{7}$$

*where the right-hand side of Eq. (7) is the max-sum Hamming distance.*

*Proof.* We first note that, for binary vectors $\boldsymbol{x}_s, \boldsymbol{x}_l \in \{0,1\}^N$, the Hamming distance is expressed as follows:

$$d_H(\boldsymbol{x}_s, \boldsymbol{x}_l) = \sum_{i=1}^{N} \left(x_{s,i}^2 + x_{l,i}^2 - 2x_{s,i}x_{l,i}\right).$$

Based on this expression, the diversity metric $\sum_{s<l} d_H(X_{:s}, X_{:l})$ can be expanded for a binary matrix $X \in \{0,1\}^{N \times S}$ as follows:

$$\begin{aligned}
\sum_{s<l} d_H(X_{:s}, X_{:l}) &= \frac{1}{2}\left(\sum_{s,l} d_H(X_{:s}, X_{:l}) - \sum_{s} d_H^2(X_{:s}, X_{:s})\right) \\
&= \frac{1}{2}\sum_{i=1}^{N}\sum_{s,l}(X_{:s,i}^2 + X_{:l,i}^2 - 2X_{:s,i}X_{:l,i}) \\
&= S\sum_{i}^{N}\left(\sum_s X_{:s,i}^2 - \frac{1}{S}\sum_{s,l} X_{:s,i}X_{:l,i}\right).
\end{aligned}$$

On the other hand, the variance of each column in a binary matrix $X$ can be expanded as follows:

$$\begin{aligned}
S^2\sum_{i=1}^{N}\mathbb{VAR}\left[\{X_{s,i}\}_{1\leq s\leq S}\right] &= S\sum_{i=1}^{N}\sum_{s'=1}^{S}\left(X_{:s',i} - \frac{1}{S}\sum_s X_{:s,i}\right)^2 \\
&= S\sum_{i=1}^{N}\sum_{s'=1}^{S}\left(X_{:s',i}^2 - 2X_{:s',i}\frac{\sum_s X_{:s,i}}{S} + \frac{\sum_{s,l} X_{:s,i}X_{:l,i}}{S^2}\right) \\
&= S\sum_{i=1}^{N}\left(\sum_{s'} X_{:s',i}^2 - \frac{2\sum_{s',s} X_{:s',i}X_{:s,i}}{S} + \frac{\sum_{s,l} X_{:s,i}X_{:l,i}}{S}\right) \\
&= S\sum_{i=1}^{N}\left(\sum_{s'} X_{:s',i}^2 + \frac{1}{S}\sum_{sl} X_{:s,i}X_{:l,i}\right) \\
&= \sum_{s<l} d_H(X_{:s}, X_{:l}).
\end{aligned}$$

By this, we finish the proof. □

# B  Additional Implementation Details

## B.1  Graph Neural Networks

A graph neural network (GNN) (Gilmer et al., 2017; Scarselli et al., 2008) is a specialized NN for representation learning of graph-structured data. GNNs learn a vectorial representation of each node through two steps. (I) Aggregate step: This step employs a permutation-invariant function to generate an aggregated

node feature. (II) Combine step: Subsequently, the aggregated node feature is passed through a trainable layer to generate a node embedding, known as 'message passing' or 'readout phase.' Formally, for given graph $G = (V, E)$, where each node feature $\boldsymbol{h}_v^0 \in \mathbb{R}^{N^0}$ is attached to each node $v \in V$, the GNN iteratively updates the following two steps. First, the aggregate step at each $k$-th layer is defined by

$$\boldsymbol{a}_v^k = \text{Aggregate}_\theta^k \left( \{ h_u^{k-1}, \forall u \in \mathcal{N}_v \} \right),$$

where the neighborhood of $v \in V$ is denoted as $\mathcal{N}_v = \{ u \in V \mid (v, u) \in E \}$, $\boldsymbol{h}_u^{k-1}$ is the node feature of neighborhood, and $\boldsymbol{a}_v^k$ is the aggregated node feature of the neighborhood. Second, the combined step at each $k$-th layer is defined by

$$\boldsymbol{h}_v^k = \text{Combine}_\theta^k(\boldsymbol{h}_v^{k-1}, \boldsymbol{a}_v^k),$$

where $\boldsymbol{h}_v^k \in \mathbb{R}^{N^k}$ denotes the node representation at $k$-th layer. The total number of layers, $K$, and the intermediate vector dimension, $N^k$, are empirically determined hyperparameters. Although numerous implementations for GNN architectures have been proposed, the most basic and widely used GNN architecture is a graph convolutional network (GCN) (Scarselli et al., 2008) given by

$$\boldsymbol{h}_v^k = \sigma \left( W^k \sum_{u \in \mathcal{N}(v)} \frac{\boldsymbol{h}_u^{k-1}}{|\mathcal{N}(v)|} + B^k \boldsymbol{h}_v^{k-1} \right),$$

where $W^k$ and $B^k$ are trainable parameters, $|\mathcal{N}(v)|$ serves as normalization factor, and $\sigma : \mathbb{R}^{N^k} \to \mathbb{R}^{N^k}$ is some component-wise nonlinear activation function such as sigmoid or ReLU function.

## C    Experiment Details

This section describes the details of the experiments .

### C.1    Architecture of GNNs

We describe the details of the GNN architectures used in our numerical experiments. For each node $v \in V$, the first convolutional layer takes a node embedding vectors, $\boldsymbol{h}_{v,\boldsymbol{\theta}}^0$ for each node, yielding feature vectors $\boldsymbol{h}_{v,\boldsymbol{\theta}}^1 \in \mathbb{R}^{H_1}$. Then, the ReLU function is used as a component-wise nonlinear transformation. The second convolutional layer takes the feature vector, $\boldsymbol{h}_{\boldsymbol{\theta}}^1$, as input, producing a feature vector $\boldsymbol{h}_{v,\boldsymbol{\theta}}^2 \in \mathbb{R}^S$. Finally, a sigmoid function is applied to the vector $\boldsymbol{h}_{\boldsymbol{\theta}}^2$, producing the higher-order array solutions $P_{v:,\boldsymbol{\theta}} \in [0, 1]^{N \times S}$. Here, for MIS and MaxCut problems, we set $|H_0| = \text{int}(N^{0.8})$ as in Schuetz et al. (2022a); Ichikawa (2024), and for the DBM problems, we set it to 2,500. Across all problems, we set $H_1 = H_0$, and $H_2 = S$. We conducted all experiments by using V100GPU.

### C.2    Training setting and post-rounding method

We use the AdamW (Kingma & Ba, 2014) optimizer with a learning rate as $\eta = 10^{-4}$ and weight decay as $10^{-2}$. The training the GNNs conducted for a duration of up to $5 \times 10^4$ epochs with early stopping, which monitors the summarized loss function $\sum_{s=1}^S \hat{l}(P_{:,s})$ and penalty term $\Phi(P; \gamma, \alpha)$ with tolerance $10^{-5}$ and patience $10^3$. After the training phase, we apply projection heuristics to round the obtained soft solutions back to discrete solutions using simple projection, where for all $i \in [N], s \in [S]$, we map $P_{\theta,i,s}$ to 0 if $P_{\theta,i,s} \leq 0.5$ and $P_{\theta,i,s}$ to 1 if $P_{\theta,i,s} > 0.5$. Note that due to the annealing, CPRA-PI-GNN solver ensures that the soft solution are nearly binary for all benchmarks, making them robust against the threshold 0.5 in our experiments.

### C.3    Problem specification

**Maximum independent set problems**    There are some theoretical results for MIS problems on RRGs with the node degree set to $d$, where each node is connected to exactly $d$ other nodes. The MIS problem is a fundamental NP-hard problem (Karp, 2010) defined as follows. Given an undirected graph $G(V, E)$, an

Table 1: The objective functions for the three problems to be studied.

| | Objective Function | Parameters |
|---|---|---|
| MIS | $l(\boldsymbol{x}; G, \lambda) = -\sum_{i \in V} x_i + \lambda \sum_{(i,j) \in E} x_i x_j$ | $|V| = N$ |
| MaxCut | $l(\boldsymbol{x}; G) = \sum_{i<j} A_{ij}(2x_i x_j - x_i - x_j)$ | $A \in \mathbb{R}^{N \times N}$ |
| DBM | $l(\boldsymbol{x}; C = \{A, M\}, \boldsymbol{\lambda}) = -\sum_{ij} A_{ij} x_{ij} + \lambda_1 \sum_i \mathrm{ReLU}(\sum_j x_{ij} - 1)$ $+ \lambda_2 \sum_j \mathrm{ReLU}(\sum_i x_{ij} - 1) + \lambda_3 \mathrm{ReLU}(p \sum_{ij} x_{ij} - \sum_{ij} M_{ij} x_{ij})$ $+ \lambda_4 \mathrm{ReLU}(q \sum_{ij} x_{ij} - \sum_{ij}(1 - M_{ij}) x_{ij})$ | $A \in \mathbb{R}^{N_1 \times N_2}$ $M \in \mathbb{R}^{N_1 \times N_2}$ $p, q \in \mathbb{R}$ |

independent set (IS) is a subset of nodes $\mathcal{I} \in V$ where any two nodes in the set are not adjacent. The MIS problem attempts to find the largest IS, which is denoted $\mathcal{I}^*$. In this study, $\rho$ denotes the IS density, where $\rho = |\mathcal{I}|/|V|$. To formulate the problem, a binary variable $x_i$ is assigned to each node $i \in V$. Then the MIS problem is formulated as follows:

$$f(\boldsymbol{x}; G, \lambda) = -\sum_{i \in V} x_i + \lambda \sum_{(i,j) \in E} x_i x_j,$$

where the first term attempts to maximize the number of nodes assigned 1, and the second term penalizes the adjacent nodes marked 1 according to the penalty parameter $\lambda$. In our numerical experiments, we set $\lambda = 2$, following Schuetz et al. (2022a), no violation is observed as in (Schuetz et al., 2022a). First, for every $d$, a specific value $\rho_d^*$, which is dependent on only the degree $d$, exists such that the independent set density $|\mathcal{I}^*|/|V|$ converges to $\rho_d^*$ with a high probability as $N$ approaches infinity (Bayati et al., 2010). Second, a statistical mechanical analysis provides the typical MIS density $\rho_d^{\mathrm{Theory}}$, and we clarify that for $d > 16$, the solution space of $\mathcal{I}$ undergoes a clustering transition, which is associated with hardness in sampling (Barbier et al., 2013) because the clustering is likely to create relevant barriers that affect any algorithm searching for the MIS $\mathcal{I}^*$. Finally, the hardness is supported by analytical results in a large $d$ limit, which indicates that, while the maximum independent set density is known to have density $\rho_{d \to \infty}^* = 2\log(d)/d$, to the best of our knowledge, there is no known algorithm that can find an independent set density exceeding $\rho_{d \to \infty}^{\mathrm{alg}} = \log(d)/d$ (Coja-Oghlan & Efthymiou, 2015).

**Diverse bipartite matching (DBM) problems** We adopt this CO problem from Ferber et al. (2020); Mulamba et al. (2020); Mandi et al. (2022) as a practical example. The topologies are sourced from the CORA citation network (Sen et al., 2008), where each node signifying a scientific publication, is characterized by 1,433 bag-of-words features, and the edges represents represents the likelihood of citation links. Mandi et al. (2022) focused on disjoint topologies, creating 27 distinct instances. Each instance is composed of 100 nodes, categorised into two group of 50 nodes, labeled $N_1$ and $N_2$. The objective of DBM problems is to find the maximum matching under diversity constraints for similar and different fields. It is formulated as follows:

$$l(\boldsymbol{x}; C, M, \boldsymbol{\lambda}) = -\sum_{ij} C_{ij} x_{ij} + \lambda_1 \sum_{i=1}^{N_1} \mathrm{ReLU}\Big(\sum_{j=1}^{N_2} x_{ij} - 1\Big) + \lambda_2 \sum_{j=1}^{N_2} \mathrm{ReLU}\Big(\sum_{i=1}^{N_1} x_{ij} - 1\Big)$$

$$+ \lambda_3 \mathrm{ReLU}\Big(p \sum_{ij} x_{ij} - \sum_{ij} M_{ij} x_{ij}\Big) + \lambda_4 \mathrm{ReLU}\Big(q \sum_{ij} x_{ij} - \sum_{ij}(1 - M_{ij}) x_{ij}\Big),$$

where a reward matrix $C \in \mathbb{R}^{N_1 \times N_2}$ indicates the likelihood of a link between each node pair, for all $i, j$, $M_{ij}$ is assigned 0 if articles $i$ and $j$ belong to the same field, or 1 if they don't. The parameters $p, q \in [0, 1]$ represent the probability of pairs being in the same field and in different fields, respectively. Following (Mandi et al., 2022), we examine two variations of this problem: Matching-1 and Matching-2, characterized by $p$ and $q$ values of 25% and 5%.

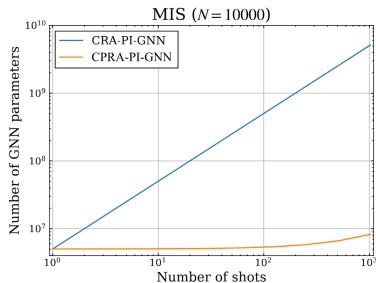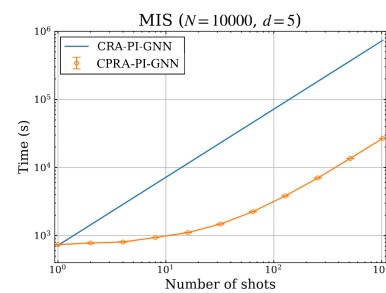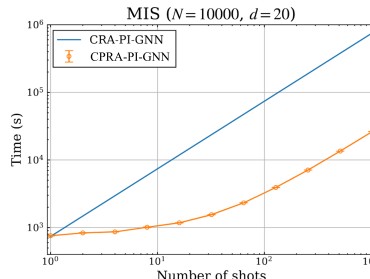

Figure 7: The runtime of the CPRA-PI-GNN solver, compared to $S$ individual runs of the CRA-PI-GNN solver, as a function of number of shots $S$. Error bars represent the standard deviations of 5 random seeds.

**Maximum cut problems** The MaxCut problem, a well-known NP hard problems (Karp, 2010), has practical application in machine scheduling (Alidaee et al., 1994), image recognition (Neven et al., 2008) and electronic circuit layout design (Deza & Laurent, 1994). It is defined as follows: In an undirected graph $G = (V, E)$, a cut set $\mathcal{C} \in E$, which is a subset of edges, divides the nodes into two groups $(V_1, V_2 \mid V_1 \cup V_2 = V, V_1 \cap V_2 = \emptyset)$. The objective the MaxCut problem is to find the largest cut set. To formulate this problem, each node is assigned a binary variable: $x_i = 1$ signifies that node $i$ is in $V_1$, while $x_i = 0$ indicates node $i$ is in $V_2$. For an edge $(i, j)$, $x_i + x_j - 2x_ix_j = 1$ is true if $(i, j) \in \mathcal{C}$; otherwise, it equal 0. This leads to the following objective function:

$$l(\boldsymbol{x}; G) = \sum_{i<j} A_{ij}(2x_ix_j - x_i - x_j)$$

where $A_{ij}$ is the adjacency matrix, where $A_{ij} = 0$ signifies the absence of an edge, and $A_{ij} > 0$ indicates a connecting edge. Following Schuetz et al. (2022a); Ichikawa (2024), this experiments employ seven instances from Gset dataset (Ye, 2003), recognized as a standard MaxCut benchmark. These seven instances are defined on distinct graphs, including Erdös-Renyi graphs with uniform edge probability, graphs with gradually decaying connectivity from 1 to $N$, 4-regular toroidal graphs, and one of the largest instance with 10,000 nodes.

# D Additional Experiments

## D.1 Runtime and # Params as a function of number of shots

In this section, we investigate the runtime of CPRA-PI-GNN solver as a function of the number of shots, $S$, compared to the runtime for $S$ individual runs of CRA-PI-GNN solver. Fig. 7 shows each runtime as a function of the number of shots $S$. For this analysis, we incrementally increase the number of shots, further dividing the range of penalty parameters from $2^{-2}$ to $2^{17}$. The results indicate that CPRA-PI-GNN solver can find penalty-diversified solutions within a runtime nearly identical to that of a single run of CRA-PI-GNN solver for shot numbers $S$ from $2^0$ to $2^{10}$. However, for $S > 10^2$, we observe a linear increase in runtime as the number of shots $S$ grows because of the limitation of memory of GPUs. Fig. 8 (right) shows the distribution of Hamming distances combination, $\{d_H(P_{:s}, P_{:l})\}_{1 \le s < l \le 300}$, and the count of unique solutions with different $\nu = 0.00, 0.05, 0.10, 0.20$, whereas Fig. 7 (right) shows the maximum ApR, i.e., $\max_{s=1,\ldots,300} \mathrm{ApR}(P_{:,s})$ as a function of the parameter $\nu$. These results indicate that the CPRA-PI-GNN solver can find more variation-diversified solutions as the parameter $\nu$ increases. Furthermore, This result indicates that the CPRA-PI-GNN solver can boost the exploration capabilities of the CRA-PI-GNN solver, leading to the discovery of better solutions.

## D.2 Additional results of variation-diversified solutions for MIS

Fig. 8 (right) shows the distribution of Hamming distances combination, $\{d_H(P_{:s}, P_{:l})\}_{1 \le s < l \le 300}$, and the count of unique solutions with different $\nu = 0.00, 0.05, 0.10, 0.20$, whereas Fig. 8 (right) shows the maximum

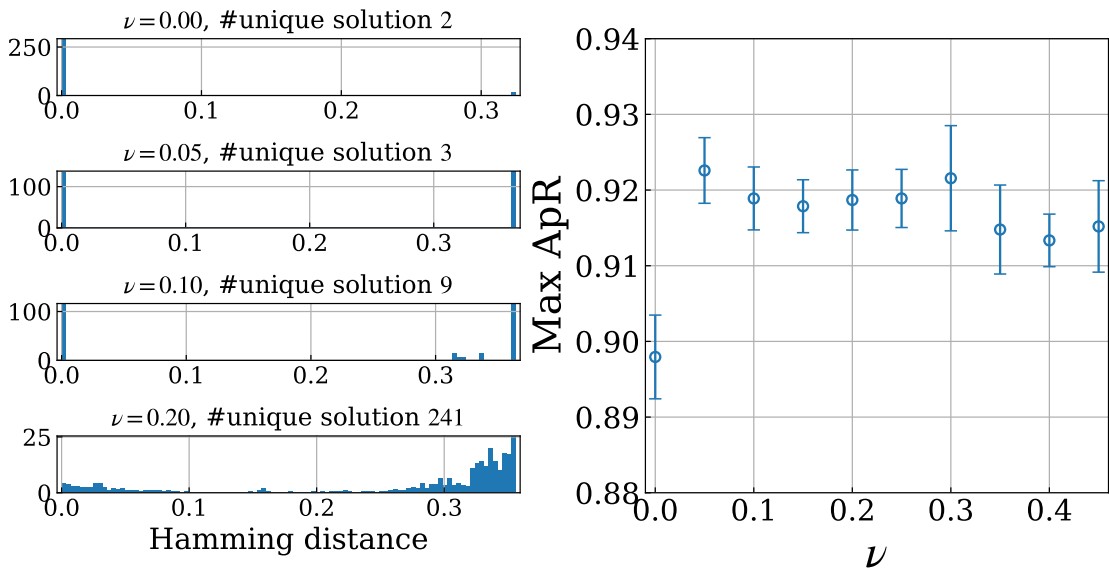

Figure 8: The density of Hamming distance combination of the solution, $\{d_H(P_{:s}, P_{:l})\}_{1 \leq s < l \leq 300}$, with different parameters $\nu$ and the count of unique solutions (left), and the maximum ApR, $\max_{s=1,\ldots,300} \mathrm{ApR}(P_{:s})$, as a function of the parameter $\nu$.

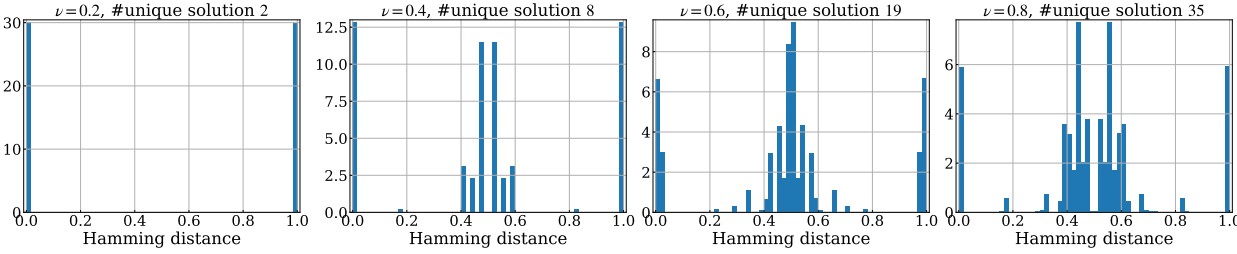

Figure 9: The density of Hamming distance combination of the solution, $\{d_H(P_{:s}, P_{:l})\}_{1 \leq s < l \leq 1000}$ in MaxCut G14, with different parameters $\nu$ and the count of unique solutions

ApR, i.e., $\max_{s=1,\ldots,300} \mathrm{ApR}(P_{:,s})$ as a function of the parameter $\nu$. These results indicate that the CPRA-PI-GNN solver can find more variation-diversified solutions as the parameter $\nu$ increases. Furthermore, This result indicates that the CPRA-PI-GNN solver can boost the exploration capabilities of the CRA-PI-GNN solver, leading to the discovery of better solutions.

### D.3 Additional results of variation-diversified solutions for MaxCut G14.

In this section, to supplement the results of the variation-diversified solutions for MaxCut G14 in Section 6.3, we present the results of the Hamming distance distribution. Fig. 9 shows the distribution of combinations of solution Hamming distances under the same settings as in Section 6.1. From these results, it is evident that the CPRA-PI-GNN solver has acquired solutions in four distinct clusters.

### D.4 Additional results for validation of exploration ability

These improvement is consistent across other Gset instances on distict graphs with varying nodes, as shown in Table. 2. In these experiment, we fix as $\nu = 6$ and evaluate the maximum ApR, $\max_{s=1,\ldots,1000} \mathrm{ApR}(P_{:s})$. This result shows that CPRA-PI-GNN solver outperforme CRA-PI-GNN, PI-GNN, and RUN-CSP solvers.

Table 2: Numerical results for MaxCut on Gset instances

| (Nodes, Edges) | CSP | PI | CRA | CPRA |
|---|---|---|---|---|
| G14 (800, 4,694) | 0.960 | 0.988 | 0.994 | 0.997 |
| G15 (800, 4,661) | 0.960 | 0.980 | 0.992 | 0.995 |
| G22 (2,000, 19,990) | 0.975 | 0.987 | 0.998 | 0.999 |
| G49 (3,000, 6,000) | **1.000** | 0.986 | **1.000** | **1.000** |
| G50 (3,000, 6,000) | **1.000** | 0.990 | **1.000** | **1.000** |
| G55 (5,000, 12,468) | 0.982 | 0.983 | 0.991 | 0.994 |
| G70 (10,000, 9,999) | – | 0.982 | 0.992 | 0.997 |

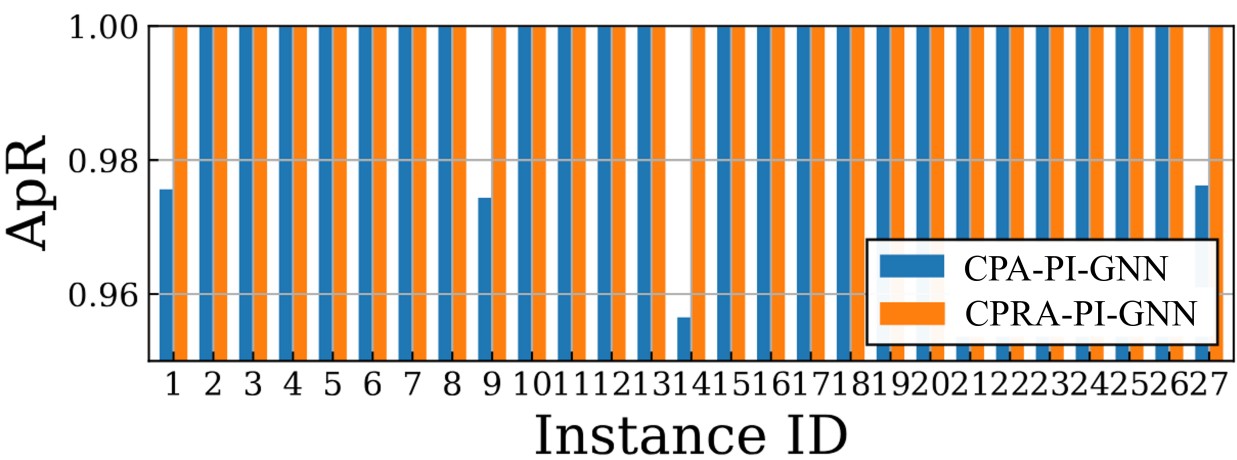

Figure 10: The ApR of DBM (Matching-1) using CPRA-PI-GNN and CRA-PI-solvers (Ichikawa, 2024).

## D.5 CPRA for Multi-instance Solutions

In this section, we numerically demonstrate that the CPRA-PI-GNN solver can efficiently solve multiple problems with similar structures. The numerical experiments solve all 27 DBM instances using the CPRA-PI-GNN solver with the following loss function:

$$\hat{R}(\boldsymbol{\theta}; \mathcal{C}_S, \boldsymbol{\lambda}, \gamma) = \sum_{s=1}^{S} \hat{l}(\boldsymbol{\theta}; C_s, \boldsymbol{\lambda}_s) + S(\boldsymbol{\theta}; \mathcal{C}_S, \gamma, \alpha), \quad S(\boldsymbol{\theta}, \mathcal{C}_S, \gamma, \alpha) \triangleq \gamma \sum_{i=1}^{N} \sum_{s=1}^{S} (1 - (2P_{\boldsymbol{\theta}, is}(C_s) - 1)^{\alpha}).$$

where $\mathcal{C}_S = \{C_s, M_s\}_{s=1}^{27}$ represents the instance parameters, and $\hat{l}$ is defined as follows:

$$l(\boldsymbol{x}; C, M, \boldsymbol{\lambda}) = -\sum_{i,j} C_{ij} x_{ij} + \lambda_1 \sum_i \text{ReLU}\Big(\sum_j x_{ij} - 1\Big) + \lambda_2 \sum_j \text{ReLU}\Big(\sum_i x_{ij} - 1\Big)$$
$$+ \lambda_3 \text{ReLU}\Big(p \sum_{ij} x_{ij} - \sum_{ij} M_{ij} x_{ij}\Big) + \lambda_4 \text{ReLU}\Big(q \sum_{ij} x_{ij} - \sum_{ij} (1 - M_{ij}) x_{ij}\Big),$$

where $\boldsymbol{\lambda}$ is fixed as $\boldsymbol{\lambda} = (\lambda_1, \lambda_2, \lambda_3, \lambda_4) = (2, 2, 12, 12)$. The parameters for the CPRA-PI-GNN solver is set the same as in Section 6.1. On the other hand, the CRA-PI-GNN solver repeatedly solve the 27 problems using the same settings as Ichikawa (2024). As a result, the CPRA-PI-GNN solver can explore global optimal solutions for all problems. Fig. 10 showcases the solutions yielded by both the CRA-PI-GNN and CPRA-PI-GNN solvers for the 27 Matching-1 instances. Matching-2 is excluded from this comparison, given that both solvers achieved global solutions for these instances. The CRA-PI-GNN solver, applied 27 times for Matching-1, accumulated a total runtime of 36,925±445 seconds, significantly longer than the CPRA-PI-GNN's efficient 5,617±20 seconds. For Matching-2, the CRA-PI-GNN solver required 36,816±149

Table 3: Runtime and solution quality on the 500-task, 1000-node weighted-MIS benchmark. "Pre" and "FT" denote warm-up and fine-tune phases of the two-stage schedule. The two-stage variant is not only faster than scratch training but also yields a markedly higher total weight.

| Method | Pre [s] | FT [s] | Total [s] | Avg. weight |
|---|---|---|---|---|
| Two-Stage $(200 \to 500)$ | 43.5 | 30.4 | **73.9** | **139.87** |
| Scratch-500 | — | — | 83.7 | 123.87 |

seconds, whereas the CPRA-PI-GNN solver completed its tasks in just $2{,}907 \pm 19$ seconds. The reported errors correspond to the standard deviation from five random seeds. These findings not only highlight the CPRA-PI-GNN solver's superior efficiency in solving a multitude of problems but also its ability to achieve higher Acceptance Probability Ratios (ApR) compared to the CRA-PI-GNN solver. The consistency of these advantages across different problem types warrants further investigation.

### D.6 Additional Results of penalty-diversified solutions for DBM problems

This section extends our discussion on penalty-diversified solutions for DBM problems, as introduced in Section 6.2. In these numerical experiments, we used the same $\Lambda_S$ as in Section 6.2 and executed the CPRA-PI-GNN under the same settings as in Section 6.1. As shown in Fig. 11, the CPRA-PI-GNN can acquire penalty-diversified solutions for all instances of the DBM.

## E  Two–Stage Extension Experiment

To validate the claim in Section 5 that a small warm-up subset $S' \subset S$ allows CPRA-PI-GNN to scale efficiently, we solve 500 weighted-MIS tasks on a fixed 1000-node, $d = 5$ regular graph. The first 200 weights form $S'$ (warm-up); all 500 form $S$. During warm-up the network has 200 heads and is trained for 8000 epochs with $\lambda$ annealed $-2 \to +2$, but the update stops once the entropy term falls below $10^{-4}$. The encoder is then frozen, the head is replaced by a fresh 500-way layer, and only that layer is optimised for 2000 epochs ($\lambda : 0.1 \to 0$, same stopping rule). A scratch-trained 500-head model serves as baselines.

As shown in Table 3, two-Stage finishes 11% faster than scratch yet attains 113% of the scratch objective and 143% of greedy. This confirms that learning a shared encoder on $S'$ and adapting only the output layer is both *faster* and *better* than training a large head from scratch on all tasks.

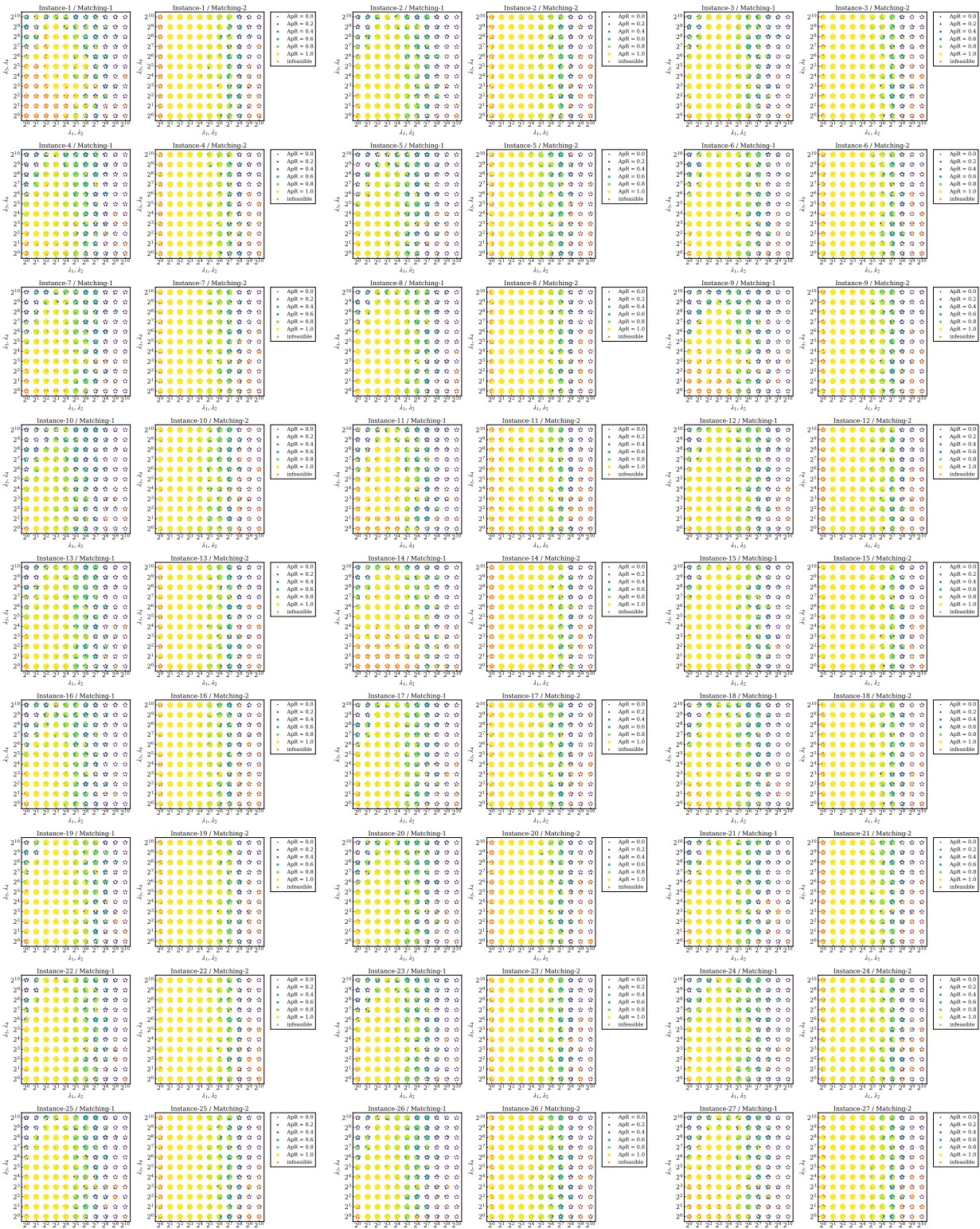

Figure 11: ApR of the DBM problems on th grid $\Lambda_S$ using CPRA-PI-GNN solver. Each point on the coordinate plane represents the results from five different random seed, with the colors indicating the ApR. The constraints violation are marked with a cross symbol.

