# OpenReview forum: "Continuous Parallel Relaxation for Finding Diverse Solutions in Combinatorial Optimization Problems"
_TMLR — Accepted by TMLR_

### Review · Reviewer_T6JK · 2025-04-12

**Summary Of Contributions:**

This paper focused on identifying diverse solutions to combinatorial optimization (CO) problems and typically aiming at scenarios where (i) hard constraints of CO are integrated into the objective functions as penalty terms with their weights being crucial and (ii) real-world cases where diversified solutions are preferable. Technically, the proposed CTRA generalized the routine of PI-GNN by extending the output from an $[n]$ tensor to a $[s, n]$ tensor, representing $s$ solutions, and tailored loss functions are applied to the two scenarios. Experiments showed that CTRA achieved comparable results to the single output version but without computing $s$ times for each value of the penalty weight, and CTRA could even improve search capabilities by considering an additional diversity penalty.

**Audience:**

Yes

**Broader Impact Concerns:**

NA.

**Claims And Evidence:**

No

**Requested Changes:**

Please see **Weaknesses**.

**Strengths And Weaknesses:**

#### Strengths
* This paper designed the idea of extending tensor dimension and leverage the GPU-based parallelization.
* The additional diversity penalty helps search for better solutions. This is interesting and makes sense since it discovers diverse local minima, but at a higher cost.

#### Weakness
* The main contribution is extending the output tensor from $[n]$ to $[s, n]$, which seems like a very minor trick. Moreover, the argument of comparable number of parameters depends on the GNNs. If we directly optimize $P$ without GNN re-parameterization, then $[n]\rightarrow [s,n]$ leads to $s$ times complexity. Only deep GNNs could amortize the $(s-1)n$ variables, but is it necessary?
* **Benchmarks**: In the first scenario (penalty-diversified), the importance of diversity is based on the assumption that the trade-off between objectives and penalties is difficult. However, it seems that such trade-off is easy for the MIS problems since most of the 20 penalty values result in zero violation.
* **Baselines**: The comparison is not clear and important baselines are missing. Is the goal to find multiple diverse solutions or to find an optimal one? If the former, then the baselines are not strong enough since they are designed for single solution. Tailored approaches for multiple solutions (e.g. [1]) should be considered; If the latter, then there are plenty of advanced ML-based solvers for COs.

[1] Li L, Aigerman N, Kim V, et al. Learning Proximal Operators to Discover Multiple Optima[C]//The Eleventh International Conference on Learning Representations.

---

> ### Author Response · Authors · 2025-05-07
> **Respond to Reviewer T6JK (1/2)**
>
> Thank you for acknowledging the strengths of our work, particularly the extension of the tensor dimension, the use of GPU-based parallelization, and the incorporation of a diversity penalty to guide the search toward better solutions.
> We address your insightful comments point by point below. For your convenience, the main revisions in the manuscript are highlighted in blue. We hope these changes satisfactorily address your concerns.
>
> > The main contribution is extending the output tensor from $[n]$ to $[s, n]$, which seems like a very minor trick. Moreover, the argument of comparable number of parameters depends on the GNNs. If we directly optimize $P$ without GNN re-parameterization, then $[n] \to [s, n]$ leads to $s$ times complexity. Only deep GNNs could amortize the $(s-1)n$ variables, but is it necessary?
>
> **Although the architectural modification is minimal, limited to expanding the final output layer, it introduces a bottleneck structure that facilitates shared representation learning across multiple instances.**
> This shared representation learning results in higher-quality solutions compared to solving each instance independently using repeated single-instance solvers, as demonstrated in Figures 3 and 10.
> The proposed diversity term reduces computational complexity from $\mathcal{O}(S^2)$ to $\mathcal{O}(S)$, substantially accelerating both forward and backward computations. These contributions are now explicitly clarified in the revised manuscript.
>
> We acknowledge that directly parameterizing $S$ solutions increases computational cost linearly with $S$, especially when each solution is modeled independently.
> Nevertheless, even under these conditions, our framework remains effective by leveraging GPU parallelism and incorporating a computationally efficient mechanism for exploring diverse solutions.
> While the optimal model complexity remains an open question for future work, this study focuses on widely used 2- or 3-layer GNNs, which are standard in UL-based CO solvers.
> Even without using deep GNNs, our shared-architecture design substantially reduces the total number of parameters compared to training $S$ fully independent models.
> As illustrated in Figure 2, our approach achieves notable parameter efficiency compared to models that produce a single solution, while maintaining or improving solution quality.
>
> > Benchmarks: In the first scenario (penalty-diversified), the importance of diversity is based on the assumption that the trade-off between objectives and penalties is difficult. However, it seems that such trade-off is easy for the MIS problems since most of the 20 penalty values result in zero violation.
>
> We appreciate the opportunity to clarify the motivation and context behind our benchmark choices.
> As the revised introduction notes, **real-world applications do not always require strictly constraint-satisfying solutions.** In some scenarios—such as scheduling with soft deadlines or resource allocation under uncertain budgets—allowing slight constraint violations can lead to significantly better overall outcomes. In such cases, offering a set of diverse solutions with different trade-offs enables end-users to make post-hoc selections based on external or context-specific criteria.
>
> While we include the Maximum Independent Set (MIS) problem as it is a widely used benchmark in neural CO solvers, our evaluation involves a more challenging and practically relevant task: Diverse Bipartite Matching (DBM). In this task, constraint satisfaction is more nuanced, as users may have heterogeneous preferences regarding diversity in matchings, making it important to provide a rich set of candidate solutions for user selection.
>
> Prior work [1, 2] has shown that even for seemingly simple problems like MIS, **the cost value can vary significantly depending on the choice of penalty coefficient $\lambda$, despite always satisfied feasibility constraints.** As a result, achieving a truly high-quality solution often requires careful and computationally expensive $\lambda$ tuning.
> Our approach mitigates this burden by producing a portfolio of solutions across a broad range of $\lambda$ values in a single training run, allowing users to achieve near-optimal objectives without repeated manual tuning or post-hoc solver calls.
>
> [1]: Haoran Sun et al, Revisiting Sampling for Combinatorial Optimization, ICML2023.
> [2]: Haoyu Peter Wang, Unsupervised Learning for Combinatorial Optimization with Principled Objective Relaxation, NeurIPS2022.

---

> ### Author Response · Authors · 2025-05-07
> **Respond to Reviewer T6JK (2/2)**
>
> > Baselines: The comparison is not clear and important baselines are missing. Is the goal to find multiple diverse solutions or to find an optimal one? If the former, then the baselines are not strong enough since they are designed for single solution. Tailored approaches for multiple solutions (e.g. [1]) should be considered; If the latter, then there are plenty of advanced ML-based solvers for COs.
>
> We emphasize that **the primary objective of this work is not to obtain a single optimal solution**, but to extend strong learning-based CO solvers—specifically CRA-PI-GNN and PI-GNN—through minimal architectural modifications: altering only the output layer.
> This lightweight modification enables efficient generation of both (i) penalty-diversified and (ii) variation-diversified solutions, with negligible additional computational overhead.
> To the best of our knowledge, no prior work has leveraged GPU acceleration and machine learning to explore penalty-diversified solutions in this way.
> **While our approach can indeed yield high-quality individual solutions, this is a secondary benefit rather than the primary focus of our study.**
>
> As noted in the reference you shared, a common limitation of reinforcement learning (RL)-based methods is a drop in performance when diversity is explicitly enforced. In contrast, our approach—similar to methods like PolyNet—achieves both diversity and solution quality simultaneously. However, unlike these methods, our goal is not to use diversity merely as a means to improve a single final solution.
> While benchmarking against diversity-enhanced RL-based methods could further strengthen our empirical evaluation, many of these approaches are based on fundamentally different paradigms—such as unsupervised learning, reinforcement learning, or supervised training—and rely on custom architectures and extensive hyperparameter tuning.
> As a result, establishing fair and standardized comparisons is often challenging. As correctly noted, conducting controlled benchmarks under aligned conditions with alternative methods remains an important direction for future work.
> Rather than exhaustively comparing with all such methods, **our goal is to demonstrate that a well-established, high-performing CO solver can be augmented in a principled and efficient manner--with only modest increases in parameter count and training time--to support both penalty- and variation-diversified solution discovery.** To further clarify our positioning, we have revised the discussion in the Related Work section.

---

> > ### Comment · Reviewer_T6JK · 2025-06-04
> >
> > As I pointed out in my previous comment: "If the former, then the baselines are not strong enough since they are designed for single solution. Tailored approaches for multiple solutions (e.g. [1]) should be considered". I do think a more comprehensive comparison study is necessary.

---

> ### Author Response · Authors · 2025-06-09
> **Reply**
>
> Thank you for your thoughtful feedback.
>
> In response, we performed an additional comparison with the multi-solution framework POL [1], concentrating on the G14 Max-Cut benchmark discussed in Section 6.3.
> To ensure a fair evaluation, we adopted the hyper-parameter settings that POL uses for its 8-node Max-Cut experiments and created a training set of $2^{14}$ samples.
> All experiments ran on an NVIDIA A100 GPU to accommodate POL's substantial memory footprint.
> Because POL's memory demand increases rapidly with problem size, we limited the number of shots to $S=100$.
>
>
> Under these settings, POL consumed
> **1 h 36 m 41 s for 50 k training epochs (Average ApR = 0.5027, Max ApR = 0.9608, DScore = 0.4982)** and
> **2 h 32 m 07s for 25 k epochs (Average ApR = 0.5068, Max ApR = 0.9576, DScore = 0.5031)**.
> These results indicate that, although POL attains a good DScore, its average solution quality is limited.
> By contrast (see Fig. 6), our approach requires no training, finishes in roughly **140 s**, and still achieves markedly better performance **(Average ApR = 0.991, Max ApR = 0.997, DScore = 0.501).**
> These findings demonstrate the superiority of our method in terms of solution quality, diversity, and computational efficiency.
>
> The original POL paper limits its experiments to small decision spaces; even there, learning the proximal operator requires between $2^{14}$ and $2^{20}$ training samples.
> Such data requirements make scaling to realistic graph sizes extremely challenging.
> Generating diverse solutions is crucial for large-scale instances where exhaustive enumeration is impossible.
> Furthermore, POL’s strategy, applying a learned proximal operator from many random initial points, does not guarantee sufficient exploration in these high-dimensional spaces; empirically, its Average ApR is lower than that achieved by simply rerunning a greedy heuristic.
>
> Although more sophisticated node encoders or a curated training set could potentially improve performance, a thorough investigation of these directions lies beyond the scope of this study and is left for future work.
>
> We would like to add further numerical results or an appendix detailing these experiments if the reviewers believe it would strengthen the manuscript.
>
> [1] Li, L., Aigerman, N., Kim, V., et al. (2023). Learning Proximal Operators to Discover Multiple Optima. ICLR 2023.

---

### Review · Reviewer_E2vj · 2025-04-15

**Summary Of Contributions:**

This paper targets a challenging and practical problem in combinatorial optimization (CO): generating diverse solutions rather than a single optimum, which is essential in scenarios involving soft constraints or approximate formulations. The authors propose a learning-based framework called Continuous Tensor Relaxation Annealing (CTRA), which extends prior UL-based solvers by incorporating parallelized representation learning over multiple instances. The framework efficiently generates:

- Penalty-diversified solutions via different penalty configurations,
- Variation-diversified solutions through a novel diversity-encouraging regularization.

The authors provide theoretical insights, clear problem formulations, and extensive numerical experiments across multiple CO tasks (MIS, MaxCut, DBM) to validate the efficiency and effectiveness of CTRA.

**Audience:**

Yes

**Claims And Evidence:**

Yes

**Requested Changes:**

1. Clarify the use of the term “tensor” in the title and paper. Since the method deals with a matrix P \in [0,1]^{N \times S} or possibly higher-order arrays, but  isn’t deeply involved with  multilinear algebra, the usage may cause confusion.
2. Highlight the main contributions more clearly in the introduction or conclusion. If the architectural novelty is limited, emphasize empirical significance or generalization ability as a key takeaway. For example, point out the real-world importance of efficiently exploring diverse solutions in large-scale or high-stakes settings.
3. Address the flexibility in inference: In Section 3.1, the loss is minimized over S instances with shared architecture. However, what happens if the number of instances S during inference differs from training? Can the model generalize to unseen S? Clarification would be valuable.
4. Clarify Proposition 3.2’s computational benefit: The proposition claims that the proposed diversity loss is more efficient than directly computing the max-sum Hamming distance. Please elaborate on detais why the computational complexity is different.

**Strengths And Weaknesses:**

**Strengths:**

+ The paper addresses a highly relevant and underexplored need for *diverse* solutions in CO.
+ The proposed CTRA framework is computationally efficient and well-suited for parallel computation on GPUs.

**Weaknesses:**

− Motivation needs refinement. While the general need for diversified solutions is explained, the connection to real-world tasks or the significance of improvement over prior solvers could be better emphasized.

− The novelty is somewhat limited in terms of technical mechanisms. The main architectural difference lies in output dimensionality and extended training across multiple instances. Although the idea is practically useful, it may be viewed as a natural extension of CRA-PI-GNN and PI-GNN.

− The notion of “tensor” in the title may mislead readers expecting tensor decomposition techniques or tensor network structures. The current usage mainly refers to a matrix or batch of solutions.

---

> ### Author Response · Authors · 2025-05-07
> **Response to Reviewer Reviewer E2vj**
>
> Thank you for your thoughtful and constructive feedback. Below, we address each of your comments in turn. In addition, major revisions to the main text are highlighted in blue for your convenience. We would appreciate it if you could kindly review them.
>
> > Clarify the use of the term "tensor" in the title and paper. Since the method deals with a matrix $P \in [0,1]^{N \times S}$ or possibly higher-order arrays, but isn’t deeply involved with multilinear algebra, the usage may cause confusion.
>
> We agree with this comment.
> Our main contribution lies in leveraging GPU-parallel continuous relaxations instead of relying on multilinear tensor algebra.
> Accordingly, we propose renaming the method to "**Continuous Parallel Relaxation (CPR)**," and will replace all occurrences of "tensor" with "higher-order array" in Sections 3 and 4.
>
> > Highlight the main contributions more clearly in the introduction or conclusion. If the architectural novelty is limited, emphasize empirical significance or generalization ability as a key takeaway. For example, point out the real-world importance of efficiently exploring diverse solutions in large-scale or high-stakes settings.
>
> We appreciate the insightful suggestion.
> We have revised the Abstract, Introduction, Related Work sections to motivate the proposed approach better.
> Although the architectural novelty is admittedly modest, we now explicitly highlight **how this simple extension, combined with instance-wise shared representation learning, enables strong UL-based CO solvers, CRA-PI-GNN and PI-GNN, to efficiently produce both (i) penalty-diversified and (ii) variation-diversified solutions.**
> We also emphasize the practical importance of such diverse solutions in real-world scenarios where soft constraints or human-in-the-loop decision-making play a critical role.
>
> > Address the flexibility in inference: In Section 3.1, the loss is minimized over $S$ instances with shared architecture. However, what happens if the number of instances $S$ during inference differs from training?  Can the model generalize to unseen $S$? Clarification would be valuable.
>
> Thank you for the thoughtful question. We apologize if our original description was unclear or caused any confusion.
> **Our method does not aim to generalize to a completely different number of instances $S$ at inference time without retraining or fine-tuning.**
> Instead, our framework enables efficient adaptation or fine-tuning when the training instance set $ \\{C_{\mu} \\}_{\mu=1}^{S^{\prime}}$ (with $S^{\prime} < S$) shares structural similarity with the full set of $S$ instances: for example, when all instances are derived from the same underlying graph and exhibit similar node degrees or neighborhood structures. In such cases, we first train the model on the initial $S^{\prime}$ instances, and then fine-tune it after expanding the output layer from size $S^{\prime}$ to $S$, while preserving the intermediate representations. This process enables the model to quickly adapt to all $S$ instances by leveraging shared representations across them.
> As shown in Figures 3 and 10, this shared bottleneck architecture, in which only the output layer differs, leads to more efficient training and often superior solution quality compared to solving each instance independently.
> Note that our method is not intended to generalize to entirely new, unseen instances by simply inputting new embedding vector at inference time. A new learning algorithm for embedding representations would be required in such cases.
> The main claim of this work, therefore, is that when a subset $S^{\prime} < S$ of structurally similar instances is available, we can pretrain on those $S^{\prime}$ instances and then fine-tune to efficiently adapt to the full set of $S$, significantly reducing computational cost.
>
> > Clarify Proposition 3.2’s computational benefit: The proposition claims that the proposed diversity loss is more efficient than directly computing the max-sum Hamming distance. Please elaborate on detais why the computational complexity is different.
>
> As shown in Proposition 3.2, the proposed diversity term can be viewed as a natural relaxation of the max-sum Hamming distance from the binary space to the continuous hypercube.
> By "natural relaxation," we mean that the proposed term coincides exactly with the max-sum Hamming distance at the vertices of the hypercube. This relaxation also offers a significant computational advantage: **whereas computing the exact max-sum Hamming distance involves all pairwise comparisons and incurs a cost of $\mathcal{O}(S^{2})$, the relaxed formulation allows for a significantly more efficient computation in $\mathcal{O}(S)$ time.** Identifying and applying this property to our method is a key part of our contribution. In the revised version, additional explanations have been added after Proposition 4.2.

---

### Review · Reviewer_6B4C · 2025-05-31

**Summary Of Contributions:**

The paper proposes CPRA, a framework for generating diverse solutions in Combinatorial Optimization (CO) efficiently. CPRA extends UL-based solvers to produce multiple solutions in parallel, improving efficiency and quality. It introduces a diversity penalty to reduce computational complexity, outperforming baseline methods in experiments.

**Audience:**

Yes

**Claims And Evidence:**

Yes

**Requested Changes:**

1. Please address the weaknesses mentioned.
2. The paper would benefit from including more discussions on some Pareto set learning works, as they are closely related (though not addressing the same problem). For example, [1] and [2]. Additionally, see the survey [3] for more relevant works.

[1] Navon and Shamsian et al., Learning the Pareto Front with Hypernetworks, ICLR 2021

[2] Lin et al., Pareto Set Learning for Neural Multi-Objective Combinatorial Optimization, ICLR 2022.

[3] Chen et al. Gradient-Based Multi-Objective Deep Learning: Algorithms, Theories, Applications, and Beyond, arXiv 2025.

**Strengths And Weaknesses:**

**Strengths**
1. The paper addresses the important real-world need for diverse solutions in combinatorial optimization.
2. It proposes a computationally efficient method for generating S diverse solutions in a single training run through output parallelization.
3. The paper is well-structured and logically presented.

**Weaknesses**
1. The proposed "two-stage learning process" (Section 5.1) for scaling to large S lacks experimental validation and further detail.
2. Lacks comparison with representative traditional (non-neural) combinatorial optimization methods.
3. No practical guidance is provided on selecting an appropriate $\lambda$ value, despite experimental results showing its impact.
4. Inconsistent naming of the method (CPRA vs. CTRA) causes confusion.

---

> ### Author Response · Authors · 2025-06-14
> **reply**
>
> Thank you for your valuable comments. We have addressed the weaknesses highlighted in your review as follows:
>
> (1) **Experimental Validation of Two-stage Learning Process**:
> We have included an experimental validation of the proposed two-stage learning process in Appendix E (Two–Stage Extension Experiment).
> This demonstrates empirically that our method achieves efficient scaling to larger $S$ without significant additional training overhead.
>
> (2) **Comparison with Traditional Greedy Solvers**:
> In Figures 5 and 6, we present the results from executing a traditional greedy solver multiple times from various random initializations. Our experiments clearly illustrate that the proposed CPRA method yields solutions with superior diversity and performance compared to random initialization.
>
> (3) **Guidance on the Penalty Coefficient $\lambda$**:
> Indeed, tuning the penalty coefficient $\lambda$ is a known challenge in constraint optimization approaches, including UL-based solvers and simulated annealing methods that utilize MCMC sampling.
> Such tuning typically demands considerable computational effort. However, our method efficiently identifies optimal $\lambda$ candidates through GPU parallelization, as demonstrated in Figures 2 and 3.
> This efficiency is a contribution to our work.
> We further argue that practical guidance on choosing $\lambda$ naturally involves user preference. Users seeking higher-quality solutions with mild constraint violations can select suitable solutions from the generated solution set during post-processing. Providing users with the flexibility to select solutions post hoc is a distinct advantage of our approach.
>
> (4) **Clarification on Method Naming**:
> We acknowledge the confusion stemming from the naming convention highlighted by Reviewer E2vj. The original term "Continuous Tensor Relaxation" (CTRA) has been reconsidered, and we propose renaming it "Continuous Parallel Relaxation" (CPR). Should this naming convention be deemed more appropriate, we will update all relevant figures and text accordingly in the camera-ready version.
>
> (5) **Expanded Discussion of Related Work**:
> We have briefly discussed the relationship of our work with pertinent Pareto set learning approaches, particularly referencing Navon and Shamsian et al. (ICLR 2021), Lin et al. (ICLR 2022), and Chen et al.'s recent survey (arXiv, 2025).

---

> > ### Comment · Reviewer_6B4C · 2025-06-16
> >
> > Thank you for your response.
> > For (2), I was referring to traditional solvers other than greedy solvers.
> > For (5), the discussion is insufficient and somewhat unclear. From my understanding, the trade-off between constraint satisfaction and cost can be interpreted as a two-objective optimization problem.

---

> > > ### Author Response · Authors · 2025-06-17
> > > **Reply**
> > >
> > > We sincerely thank the reviewer for the insightful comments.
> > >
> > > **(2) In comparison with traditional solvers**
> > >
> > > We recognize that evaluating our approach against commercial MIP solvers such as Gurobi, especially in terms of the speed–quality trade-off, would be valuable. However, the primary contribution of this work is a minimal extension of the UL-based solver CRA-PI-GNN that allows us to generate both penalty-diversified and variation-diversified solution sets efficiently on GPUs. Although commercial solvers consistently produce high-quality solutions, (i) they cannot readily exploit GPU parallelism, and (ii) they are not designed to explore solution diversity through random initialization. Consequently, we did not include them as baselines. A thorough study that injects random seeds into such solvers and evaluates the resulting speed–quality trade-off remains an important direction for future work.
> > >
> > > **(5) On clarifying the related work**
> > >
> > > In response to your feedback, we have emphasized the perspective that treats cost and constraint violation as two distinct objectives. We have also strengthened the discussion of related Pareto-front learning methods.

---

> > > > ### Comment · Reviewer_6B4C · 2025-06-22
> > > >
> > > > Thanks for your response. My concerns have been addressed, and I am inclined to accept the paper.

---

### Decision · Action_Editor_2FJt · 2025-07-19

**Recommendation:** Accept as is

**Audience:**

Yes

**Audience Explanation:**

Combinatorial optimization is an important field. Generating diverse solutions will be useful in many scenarios, especially high-stakes applications.

**Claims And Evidence:**

Yes

**Claims Explanation:**

Summary

This paper introduces an approach to efficiently generate diverse solutions for combinatorial optimization (CO) problems. To obtain S diverse solutions, the authors extend the model’s output dimension from 1 to S and introduce two types of regularization: penalty-diversified and variation-diversified regularization. The penalty-diversified regularization is a sum of S different penalty terms, and the variation-diversified regularization is a standard deviation across solutions. The paper provides theoretical justification for both regularizations. Experiments on benchmark problems, including Maximum Independent Set (MIS), MaxCut, and Diverse Bipartite Matching (DBM), demonstrate the effectiveness of the proposed method.

Strengths:

- The proposed method is simple, effective, and easy to implement. It is potentially practical for real-world applications.

- The paper includes a theoretical analysis that provides insight into the design and utility of the proposed regularization.

- Empirical results show that the approach achieves comparable or better performance to baselines, but with significantly fewer parameters and runtime.

Weaknesses:

- The experimental comparison is limited to only two baselines. Common combinatorial solvers and methods explicitly designed to generate diverse solutions are not included.


The major concern from the reviewers is the novelty of the method. Extending the output dimension and adding regularization are relatively natural ideas. According to TMLR review policy, novelty alone is not a reason for rejection. Although the methodology is simple, the paper presents a practical and well-justified solution for generating diverse solutions in CO problems, which makes a valuable contribution to the field. I encourage the authors to incorporate reviewer feedback, particularly by expanding the empirical comparison with more relevant baselines.